# Hydrogel microphones for stealthy underwater listening

Yang Gao[1,2], Jingfeng Song[3,4], Shumin Li[2,4], Christian Elowsky[5], You Zhou[5], Stephen Ducharme[3,4], Yong Mei Chen[1], Qin Zhou[2,4] & Li Tan[2,4]

Exploring the abundant resources in the ocean requires underwater acoustic detectors with a high-sensitivity reception of low-frequency sound from greater distances and zero reflections. Here we address both challenges by integrating an easily deformable network of metal nanoparticles in a hydrogel matrix for use as a cavity-free microphone. Since metal nanoparticles can be densely implanted as inclusions, and can even be arranged in coherent arrays, this microphone can detect static loads and air breezes from different angles, as well as underwater acoustic signals from 20 Hz to 3 kHz at amplitudes as low as 4 Pa. Unlike dielectric capacitors or cavity-based microphones that respond to stimuli by deforming the device in thickness directions, this hydrogel device responds with a transient modulation of electric double layers, resulting in an extraordinary sensitivity ($217\,\mathrm{nF\,kPa^{-1}}$ or $24\,\mathrm{\mu C\,N^{-1}}$ at a bias of 1.0 V) without using any signal amplification tools.

[1] State Key Laboratory for Strength and Vibration of Mechanical Structures, International Center for Applied Mechanics and School of Aerospace, Collaborative Innovation Center of Suzhou Nano Science and Technology, Xi'an Jiaotong University, Xi'an 710049, China. [2] Department of Mechanical and Materials Engineering, University of Nebraska, Lincoln 68588-0526, Nebraska, USA. [3] Department of Physics and Astronomy, University of Nebraska, Lincoln 68588-0299, Nebraska, USA. [4] Nebraska Center for Materials and Nanoscience, University of Nebraska, Lincoln 68588-0298, Nebraska, USA. [5] Center for Biotechnology, University of Nebraska, Lincoln 68588-0665, Nebraska, USA. Correspondence and requests for materials should be addressed to L.T. (email: ltan4@unl.edu) or to Q.Z. (email: zhou@unl.edu) or to Y.M.C. (email: chenym@mail.xjtu.edu.cn).

The oceans cover $\sim 71\%$ of the Earth's surface area, with only 5% being explored by human activities[1] (http://oceanservice.noaa.gov/facts/exploration.html). Towards the explorations, numerous underwater vehicles have been developed with a great amount of knowledge learned from fish. To complement vision, fish adopt a lateral line system to sense pressure variations and to detect water flows and acoustic waves[2,3]. These skills help them in mastering swimming behaviours such as rheotaxis, schooling and prey tracking[4,5]. Similarly, underwater vehicles, such as submarines, monitor flow velocities and sound waves to navigate, identify hostile objects, track ocean currents and surface waves, and communicate with each other[6,7]. However, in the current era of stealthy warfare, conventional ceramic piezoelectric sound navigation and ranging (SONAR) systems[8,9] suffer from large acoustic impedance mismatch with water, rendering them easily detectable, because they efficiently reflect incoming acoustic signals. For example, a piezoelectric ceramic such as PZT has a density ($\rho$) of $7,600 \, kg \, m^{-3}$ and a bulk modulus ($K$) close to 100 GPa (refs 10,11). In comparison, water has much smaller values in both ($\rho = 1,000 \, kg \, m^{-3}$ and $K = 2.0 \, GPa$)[12]. Therefore, the acoustic impedance ($\sqrt{\rho K}$) of PZT is $> 20$ times that of water! This large mismatch introduces two problems. It reflects 80% incoming acoustic power, reducing detectable signal fivefold, and returning a large echo that makes the sensor highly detectable by scanning SONAR. In addition, the detection efficiency of PZT-based acoustic sensors is relatively poor at low frequencies. Alternatively, suspended thin membranes of poly(vinylidene fluoride)[13] or graphene[14] stretched over air cavities have been proposed as microphones to afford a higher sensitivity than PZT[15], or to detect ultrasound from bats, but these configurations introduce even larger mismatch in acoustic impedance between the device (air) and water. Considering the advances in acoustic metamaterial cloaking[16,17], which greatly attenuates incoming acoustic signals, thereby concealing submarine bodies from SONAR detection, a ceramic piezoelectric detector or cavity-based microphone, which is necessarily kept outside of this 'invisibility cloak', remains a strong acoustic reflector.

In contrast to a rigid solid such as ceramics or a low-density compliant medium such as air, hydrogels are soft polymeric materials that, being mostly water, have almost perfect acoustic impedance matching with water[18]. Polar functional groups from the backbone or side chains allow hydrogels to absorb a large amount of liquid into a three-dimensional polymer networks without leaking. Unlike a dielectric capacitor, where the capacitance is governed by the distance between two parallel electrodes[19–22], hydrogel capacitors derive their capacitance from electrical double layers (EDLs)[23,24]. When a piece of gel is sandwiched between two electrodes and when the electrodes are biased, a thin layer of charged ions will be attracted to the nanometre vicinity of the electrodes. The gap between this layer of ions and that opposite charges from the electrodes defines the thickness, or Debye length ($\kappa^{-1}$)[25], of the EDL. Due to a small value of $\kappa^{-1}$, capacitors based on this mechanism have three to five orders of magnitude higher specific capacitance than those with purely dielectric media[21]. Furthermore, the EDL capacitors can be optimized by carefully matching the sizes of the ions to the pore sizes of the electrodes[26].

With the excellent acoustic impedance match to water, a hydrogel capacitor seems to be a promising pressure sensor or acoustic transducer. The problem, however, is that the low compressibility of water means that an EDL capacitor would have low sensitivity to pressures. In addressing this limitation, we report here that a suitable sensor can be made by incorporating a deformable network of metal nanoparticles (MNPs) into the hydrogel. The MNP network makes the capacitor highly sensitive to mechanical stimuli through a coupling between deformation of the MNP network and the ion modulation. As a result, this MNP–hydrogel capacitor is able to detect deformation, pressure and acoustic waves.

## Results

**Highlight of hydrogel microphone.** Figure 1a shows an example of a $9 \, mm^2$ hydrogel microphone fabricated by forming MNP network consisting dendritic structures 2–3 µm in size and being buried inside the soft and translucent hydrogel matrix (Fig. 1b). This MNP–hydrogel microphone was electrically biased at 1 V and submerged in water (Fig. 1c), where it picked up acoustic waves and produced a signal 30 dB stronger at low frequencies than a commercial hydrophone (Fig. 1d). Moreover, the hydrogel microphone has a wide frequency response, up to 2 kHz (Fig. 1e) and has a pronounced directional sensitivity perpendicular to the sensor surface (Fig. 1f). The slight distortion of response signals can be ascribed to the non-ideal sound transfers from the amplifier and loudspeaker to the water tank (Supplementary Fig. 1). Even more noteworthy, this device provides an estimated sensitivity of $217 \, nF \, kPa^{-1}$ ($\Delta C$ per unit pressure) for acoustic waves, $> 10,000$ times the sensitivity of conventional capacitive sensors incorporating elastic pyramids ($15.6 \, pF \, kPa^{-1}$; device area of $64 \, mm^2$)[20] and 6,000 times the sensitivity of capacitive sensors incorporating silver nanowires ($34.2 \, pF \, kPa^{-1}$; area of $16 \, mm^2$)[19]. The MNP–hydrogel capacitor also has favourable performance as an electromechanical transducer, with a response of $24 \, \mu C \, N^{-1}$, in comparison with the best high-performance piezoelectric oxides, such as lead magnesium niobate-lead titanate solid solution, which has a response of $2.8 \, nC \, N^{-1}$ (refs 27,28). Unlike David Hughes' early version of acoustic transmitter[29] (carbon microphone) that works by compacting loosely connected carbon particles between two metallic plates with a change in resistance, the hydrogel device is completely void or cavity-free, implying zero-reflection towards SONAR scanning. In the following sections, we describe the fabrication of the MNP–hydrogel capacitor and the mechanisms of coupling between capacitance and pressure.

**Fabrication of MNP–hydrogel capacitors.** Given a flat electrode, a uniform EDL will cover the entire electrode surface. If the surface is porous, then only pores larger than the EDL thickness will contribute to the effective area. If we make a porous, deformable electrode inside hydrogels, we could then tune the area of the EDL by simply changing the size of these pores. One type of such porous electrode is a network of MNPs.

The capacitor design requires two key components, an MNP network formed within the hydrogel and conducting electrodes on the surface. Producing the MNP networks presents the first challenge, because the nanoparticles are too large to diffuse into the hydrogel. Current methods for incorporating nanoparticles into hydrogels include mixing nanoparticles during the hydrogel synthesis[30] or using ultraviolet light[31] or γ-rays[32] to initiate particle formation from noble metal ions, but these methods are not suitable for making the MNP networks. The second challenge is fabricating contacts on the surface of the hydrogel after the MNP network has been formed. Conventional vacuum deposition methods for applying metal patterned electrodes to solid dielectric materials are not suitable for use with hydrogels, because water in the gels will evaporate in vacuum. Therefore, to fabricate the microphones, we turned to molecular diffusion and self-assembly to synthesize a dendritic network of MNPs directly inside the hydrogel, and to electroless plating to form the device electrodes.

The MNP network was synthesized using the following procedure, which is shown schematically in Fig. 2a. In step 1, silver ions were diffused into a slab of hydrogel from an aqueous

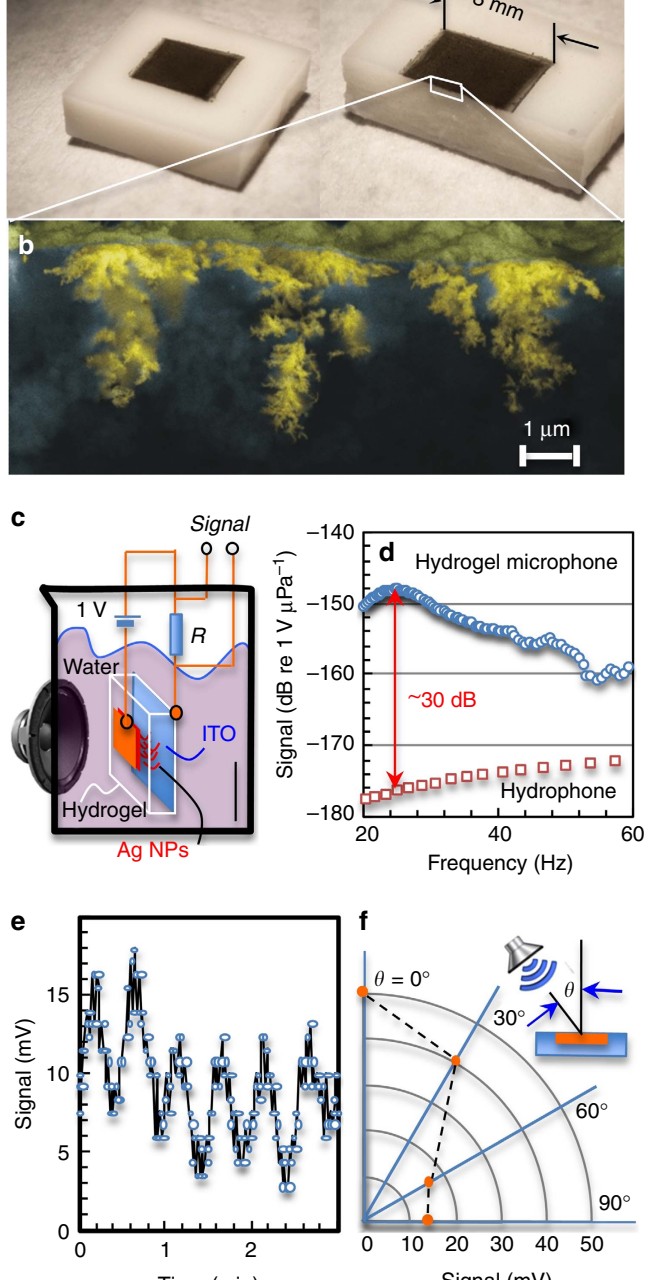

**Figure 1 | Highlight of hydrogel microphone.** (**a**) Photos (full view and sliced) and (**b**) scanning electron microscopy (SEM) image of the hydrogel membrane implanted with a patch ($3 \times 3\,mm^2$) of silver dendrites (highlighted yellow). (**c**) Set-up and circuit using the membrane as a microphone. (**d**) Better performance of the hydrogel microphone at low frequencies than a commercial device (hydrophone). (**e**) Hydrogel microphone is capable of detecting underwater sound at 2 kHz and (**f**) at all angles. Note: the 0° orientation is for top surface of the microphone facing towards the loudspeaker.

solution of silver nitrate ($AgNO_3$). Then, in step 2, the hydrogel was removed from the solution, placed on a glass slide coated with a transparent anode of indium tin oxide and covered with an amorphous silicon (a-Si) wafer to serve as a photocathode, which was biased at 3 V with respect to the anode. In step 3, the cathode was illuminated through a mask with 630 nm light with a power density of $222\,mW\,cm^{-2}$ for a total exposure of 30 s. The mask

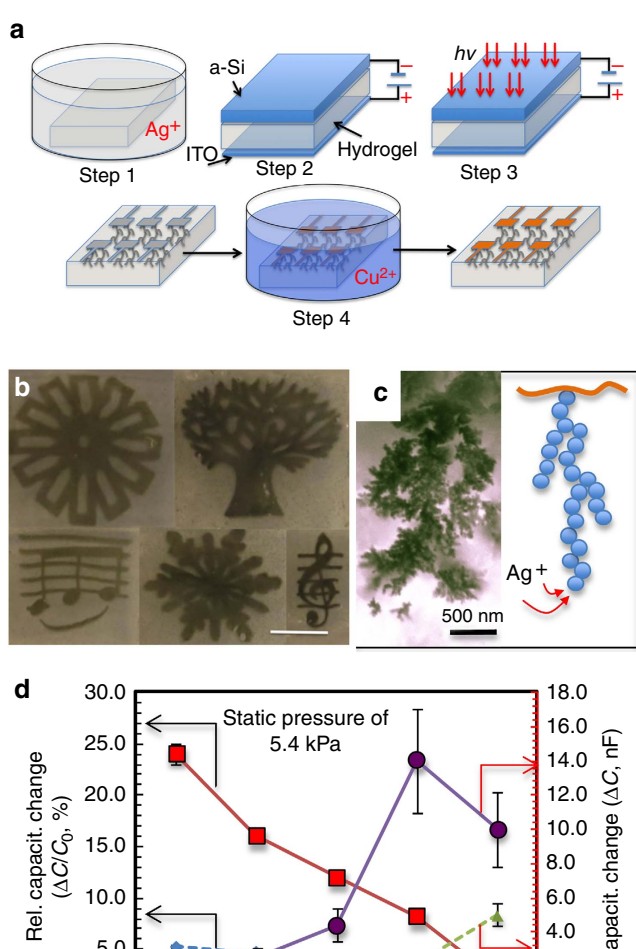

**Figure 2 | Fabrication of hydrogel microphones and their response to static pressures.** (**a**) Steps to fabricate deformable network of metal nanoparticles (MNPs) and surface electrode: (step 1) hydrogel pre-soaked in an aqueous bath of $AgNO_3$ (10 mM); (step 2) hydrogel is sandwiched and biased between an amorphous silicon (a-Si) and an ITO plate; (step 3) photo-activated a-Si reduces $Ag^+$ into $Ag^{(0)}$ nanoparticles at specific locations; (step 4) MNP–hydrogel soaked in copper sulfate bath to prepare a smooth and robust layer of surface electrodes. (**b**) Photos of patterned Ag nanoparticles in skin depth of the hydrogel. Scale bar, 3.0 mm. (**c**) High-resolution SEM image and schematic of a dendritic MNP network inside the hydrogel. (**d**) MNP–hydrogel (solid lines) responds to a static pressure of 5.4 kPa with more than four times in relative capacitance change or seven to eight times in capacitance change than MNP-free device (dashed lines). Note: data from Table 1, where tests with three samples for each data point are performed. The relative errors of $\Delta C$ and $\Delta C/C_0$ for MNP–hydrogel are, respectively, 21–25% and 3.5–4.5%. For MNP-free hydrogel, the relative errors of $\Delta C$ and $\Delta C/C_0$ are, respectively, 13.0–13.5% and 3.0–4.0% (Supplementary Table 1).

defined six independent microphones measuring $3 \times 3\,mm^2$. The areas of the insulating a-Si that were exposed to the illumination becomes photoconducting[33] allowing electrons to pass through the indium tin oxide (ITO) anode and reducing the silver ions, which arranged metallic silver nanoparticles into a dendritic network. Since the network of silver nanoparticles is organized mainly normal to the surface, they do not constitute a contiguous electrode. The ITO glass and a-Si electrodes were

removed and the hydrogel was rinsed with deionized (DI) water to remove the remainder of the silver salts.

For deposition of the electrodes by electroless plating, step 4, the hydrogel was immersed in an aqueous bath of copper sulfate $(CuSO_4 \cdot 5H_2O)$[34,35]. Because the Ag nanoparticles are highly catalytic towards copper reduction, electrodes of copper were selectively deposited on surfaces where the MNP networks were formed under illumination. Once the hydrogel surface was wiped clean, a new ITO anode was applied to the bottom as a static/dynamic load sensor. Because the MNP growth inside the hydrogel matrix occurs at the cathode side (reduction of metal ions), any metals or even metal supported a-Si can be used as the contact electrode (steps 2 and 3). In contrast, the anode (positive bias) must be expendable like aluminum or ITO to prevent water electrolysis or decomposition inside the hydrogel.

The method of forming the dendritic MNP network by selective area photoconduction affords a simple and versatile means for forming complex microphone patterns. This is illustrated in Fig. 2b, where a digital projector and optical microscope were used to illuminate the hydrogel with a variety of structures[36]. Furthermore, the method affords a convenient technique for producing customized microphone arrays (acoustic emitters) with tailored reception (emission) patterns suitable, for example, for synthetic aperture array SONAR or ultrasound[37,38] or for producing structured acoustic beams.

It is perhaps remarkable that this fabrication process works so well. One reason is that electroless plating in Fig. 2a deposits a smooth and robust layer of copper over the areas that having silver nanoparticles, creating an easy and reliable electrical access to the MNP network. However, copper growth can also start from the inside of the gel body. For example, copper ions from sulfate bath can diffuse into the hydrogel first, then adsorb on the surfaces of imbedded Ag nanoparticles before their later reduction into a copper metal. While one would think that doing the electroless plating over a short duration will limit the amount of copper ions in the hydrogel body, a kinetically fast chemical reaction usually makes the copper electrode grainy and therefore mechanically weak and fracture prone. This problem can be mitigated by increasing the crosslink density of hydrogel, which greatly impedes diffusion of the copper ions. This is because diffusion of copper ions in the hydrogel is largely controlled by the polymer networks through van der Waals[39] and electrostatic interactions[40]. We found increasing the concentration of monomers (3.0 M) during hydrogel preparation adequately prevented copper growth in the gel.

Another remarkable feature is that silver ions were not reduced into a continuous and lush silver mirror as in conventional electroplating[41,42]. This is in large part because the hydrogel is a highly viscous material[43]; as the silver ions collect inside the gel matrix, they naturally form dendritic networks (Fig. 2c; Supplementary Fig. 2). From a materials physics point of view, we can suppose that the silver growth occurs at the structural defect

sites of the gel membrane. In other words, water-swollen hydrogels can be regarded as aggregates of tiny polymer blobs containing polymer networks[44,45]. Inside each polymer blob or network, there are water molecules hydrogen bonded to the polymer network where they are also associated with silver ions. Outside the blob there are water-rich gaps that can be regarded as boundaries between these polymer blobs or simply as structural defect sites. As most water molecules inside these gaps are free to move, a quick reduction of silver ions triggers the formation of silver nanoparticles within the channels linking the blobs. In the end, the nanoparticles will be distributed deep into the hydrogel, not accumulating at the surface (highlighted as a brown wavy line, Fig. 2c). After the silver ions near the surface are reduced to form nanoparticles, they extend the cathode further into the hydrogel. This is similar to the dendritic nature of electrical breakdown in materials (or lightning in air), where plasma trails branch propagate as they extend the electrical potential surface.

**Device response to static load.** Now we will shift the focus back to the capacitive device constructed and look at their response under a static load of 5.4 kPa (Table 1; Fig. 2d). Three things are worth of noting: the first is the relatively large capacitance value (tenths of a nF) of the devices and their wide range of tunability (0.94–312.5 nF, Table 1), which is readily controlled by salt concentrations ($10^{-2}$ to 1,000 mM; see Methods for details). Unlike those dielectric counterparts[19–22,46,47], where the capacitance (pF) is determined by device thicknesses that is usually fixed at fabrication, the small EDL thickness or short Debye length[25,48] along the electrode/hydrogel interface, results in a relatively large capacitance, approximately independent of device thickness, and furthermore affords control through salt concentration. Instead of being a uniform structure, EDL consists of two parts[25,49–51]: one is the compact layer that closely sits next to the electrode surface, with a gap distance ($\delta$) of subnanometre, the other is the diffusion layer caused by irregular thermal motion of ions, which can extend to some distance away from the electrode surface. Notably, the thickness of this diffusion layer ($d$) is related to Debye length ($\kappa^{-1}$), which can range from subnanometres at high salt concentration to several hundreds of nanometres at low salt concentration[25]. In the present study, the electrolyte solution is at low molarity, where the Debye length $\kappa^{-1}$ is much larger than the compact layer gap distance $\delta$. Hence, the EDL capacitance is dominated by the diffusion length, where the thickness is usually tens to hundreds of nanometres. Certainly, when salt concentration increased by five orders of magnitude, from $10^{-2}$ to $10^3$ mM, the device capacitance increased by 300 times. Such a large capacitance also produces greater charge on deformation, potentially resulted in a large signal-to-noise ratio. Next, we examine the intriguing behaviour in Fig. 2d (dashed lines), where the control sample (MNP-free hydrogel) also exhibited an appreciable capacitance change of 0.05–5.0 nF

**Table 1 | Pressure-induced capacitance change.**

| Salt concentration (mM) | MNP-free hydrogel | | | | MNP–hydrogel | | | |
|---|---|---|---|---|---|---|---|---|
| | $C_0$ (nF) | $\Delta C$ (nF) | $\Delta C/C_0$ (%) | $\Delta n_\infty/n_\infty$ (%) | $C_0$ (nF) | $\Delta C$ (nF) | $\Delta C/C_0$ (%) | $\Delta A/A$ (%) |
| $10^{-2}$ | 0.94 | 0.05 | 5.3 | 10.9 | 3.45 | 0.83 | 24.0 | 17.7 |
| 1 | 6.38 | 0.3 | 4.7 | 9.6 | 15.625 | 2.5 | 16.0 | 10.8 |
| 10 | 15 | 0.6 | 4.0 | 8.2 | 36.67 | 4.4 | 12.0 | 7.7 |
| 100 | 81.5 | 2.2 | 2.7 | 5.5 | 170.73 | 14 | 8.2 | 5.4 |
| 1,000 | 312.5 | 5.0 | 1.6 | 3.2 | 476.2 | 10 | 2.1 | 0.5 |

Note: all the values of $C_0$, $\Delta C$ and $\Delta C/C_0$ showed here are the average values using three samples by performing one test on each sample.
Ion concentration modulation ($\Delta n_\infty/n_\infty$) and deformable MNP ($\Delta A/A$) in hydrogel capacitors contribute to capacitance variation under a static pressure of 5.4 kPa.

under pressure, for a nontrivial relative capacitance change ($\Delta C/C_0 = 1.6$–$5.3\%$). If one considers that the aluminum electrode (foil) did not form a conformal contact to the control sample, an increase in electrode contact area on the elimination of voids between the foil and the hydrogel could cause an increase in capacitance. However, such an increase shall not be inversely proportional to salt concentrations. A decrease in EDL thickness can, on the other hand, increase the capacitance; but expecting a moderate pressure of 5.4 kPa to directly shorten the distance between the ion–electron pairs in a planer EDL is unrealistic. Rather, a local variation in ion or salt concentration along the electrolyte–electrode interface is a more likely mechanism. In fact, hydrogels such as polyacrylamide (PAAm) have been known for their structural heterogeneity[52,53], where polymer networks or blobs of different sizes individually control its ion retention and mobility. Such that, under compression, deformed hydrogel networks could release trapped ions to both electrodes. These extra ions can then alter the Debye length of the EDL, resulting in an increased capacitance. Time wise, a relocation of ions over a distance of 10 μm ($L$) would take $\sim 50$ ms ($\tau = L^2/2D$, assuming diffusion constant ($D$)[54] of $10^{-5}$ cm$^2$ s$^{-1}$), suggesting a fast enough response to a static or low-frequency load.

Analytically, the capacitance of the device (Al/hydrogel/ITO) is dominated by the EDL capacitor on the aluminum side, as it is 25 times smaller in area than the ITO plate (see Methods). This EDL capacitance can be estimated using a simple capacitor model, that is, $C_0 = \varepsilon\varepsilon_0 \frac{A}{\kappa^{-1}}$, where $\varepsilon$ is the relative dielectric constant and $\varepsilon_0$ is the electric permittivity of the vacuum, $A$ is the effective area of the electrode, and $\kappa^{-1}$ is the EDL thickness. Since EDL thickness is inversely proportional to the square root of ion concentrations ($n_\infty$)[25], the capacitance change due to ion concentration change ($\Delta n_\infty$) is:

$$\Delta C = \varepsilon\varepsilon_0 \frac{A}{\kappa^{-1}} \left( \frac{\kappa^{-1}}{\kappa^{-1\prime}} - 1 \right) = C_0 \left( \sqrt{1 + \frac{\Delta n_\infty}{n_\infty}} - 1 \right)$$
$$\approx \frac{1}{2} C_0 \frac{\Delta n_\infty}{n_\infty}. \qquad (1)$$

A linear relationship between relative capacitance change and ion concentration change can therefore be established (Table 1). Under compression, the MNP-free hydrogel experiences a moderate concentration fluctuation ($\Delta n_\infty/n_\infty = 10.9$–$3.2\%$) at the electrolyte–electrode interface. These changes contribute to a reduced EDL thickness and a large perturbation (5.3%) in capacitance for devices having less salt ($10^{-2}$ mM) or a small perturbation (1.6%) to those having more salt ($10^3$ mM). Even though the latter perturbation is small, a large initial capacitance (312.5 nF) in a stress-free device yet manifests into a nontrivial increase of 5.0 nF (Fig. 2d).

The implantation of MNPs presents a much more effective way to tune the overall capacitance, while the ion concentration modulation ($\Delta n_\infty$) provides further control even with already-fabricated devices. The distortion of the dendritic or fractal network of MNPs directly changes the effective electrode area ($\Delta A$). Indeed, under the same pressure, we saw approximately three to four times of increased response in relative capacitance change ($\Delta C/C_0 = 8.2$–$24\%$) or seven to eight times of response in capacitance change ($\Delta C = 0.83$–$14.0$ nF; solid lines, Fig. 2d). Since $\Delta C$ is dependent on both $\Delta n_\infty$ and $\Delta A$, if we assume ion concentration varies the same amount under the same pressure variation (that is, $\Delta n_\infty$ is the same), then we can extract $\Delta A$ directly from the device response (Supplementary Note 1). As shown in Table 1, under compression, the deformable network of MNPs experiences a $\sim 18\%$ increase in effective electrode area at the lowest salt concentration ($10^{-2}$ mM), but a moderate increase (5%) at an intermediate salt condition of 100 mM. To better appreciate this variation in response, let us take a closer look at the diagrams in Fig. 3a and pay attention to the non-uniform spacing between nanoparticles inside this spongy network structure. Following the dendritic shape of MNP network (Fig. 2c), metal particles at the end of those branches will be spaced farther away from the stem than those near the stem–branch joints. Now, when a vertical load is applied at the top surface (compression mode, Fig. 3a), the structure of the MNP branches along the vertical direction (the stem) start to separate from those at slanted angles (branch). As a consequence, the opening between the branch and stem will increase, so that each branch releases more 'free' spaces for the formation of additional double layers. However, a thick EDL (low ion concentration) can only be inserted in those wide gaps at the far end of branches; a thin EDL (high ion concentration), on the other hand, can be tucked close into the stem–branch joints. If true, this model of a concentration-dependent response can be used to explain the sudden drop for $\Delta C$ shown in Fig. 2d and it also implies that slender branches on the side are the main sites for EDL tuning. In contrast, when the MNP network is pushed from the side (shear mode, Fig. 3a), gaps between the stem–branches will close. As such, few ions will be trapped for EDL build-up. Either way, a capacitance increase or decrease should be observed, depending on the location of external loads relative to the sites of MNP branches. As shown in Fig. 3b, we did find good agreement with above analysis, where arrays of hydrogel capacitors show increased capacitance over areas having a direct contact with the external loads. Other sections further away from the peak loads show diminished values.

We use the data shown in Fig. 3c to verify that shear force is indeed the main cause to capacitance decrease. In this experiment, an airflow of 50 ml s$^{-1}$ was incident on the MNP–hydrogel surfaces from four different angles (30–90°; $\sim 1.0$ kPa in pressure at $\theta = 90°$). Regardless of the airflow directions, all four devices exhibit reduced capacitance, with greater reduction at larger incidence angles. This might seem counter-intuitive, but it is easily understood by noting that the air streamlines conform to the hydrogel surface and mainly induces shear force. Parallel incidence allows longer distances to develop thicker boundary layers, which reduces the velocity gradient near the hydrogel surface, and thus the shear force is smaller[55]. Yet, one seemingly undesired feature observed in Fig. 3c is the long recovery time ($\sim 5.0$ s) after turning off the airflow, whereas the loading stage is much faster (0.1 s). Since the external load is applied over the bulk of hydrogel device, with gel as the major mechanical structure of the device, the device response follows the viscoelastic behaviour of the gel. At the beginning of the loading step, the gel is far away from the new equilibrium state and hence deforms quickly towards it. However, the loading step is very short, and when the stress is released, the gel is only slightly perturbed from its original equilibrium state, and therefore relaxes slowly towards it. The observed slow recovery time is due to the small equivalent spring constant (Young's modulus)[56,57] or ultra-soft nature of the hydrogel membrane. Because the MNP network is embedded in the gel matrix, how fast the pore can be opened or closed critically depends on the response of the gel matrix. Overall, while compressive forces induce capacitance increase for the device, we found that shear force can indeed induce a capacitance decrease.

**Device response to underwater acoustic waves.** The high sensitivity of EDL capacitance to mechanical deformation (Supplementary Fig. 3) encourages us to explore its potential application as an underwater microphone, where the mechanical

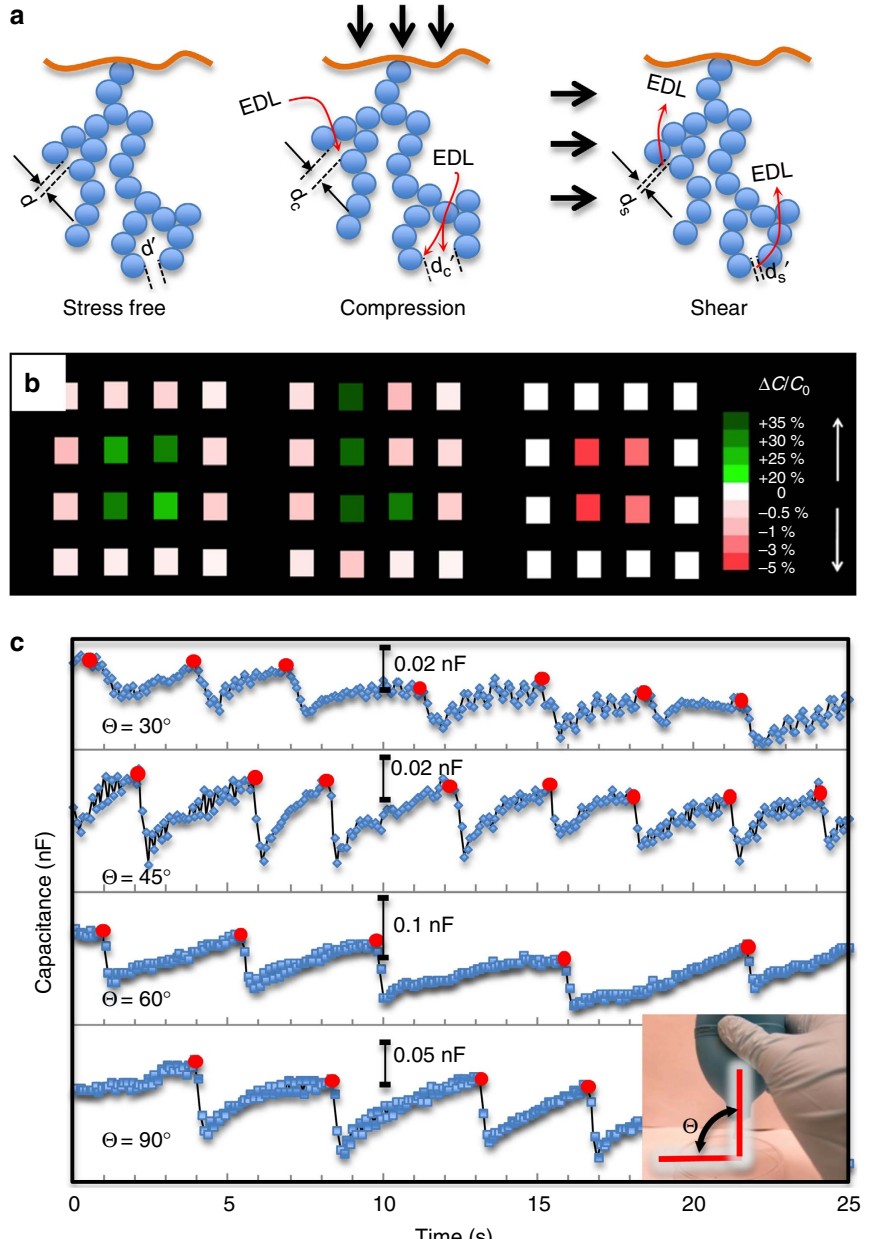

**Figure 3 | Response of hydrogel capacitors to deformations in air. (a)** Deformable network of metal nanoparticles (MNPs) makes the hydrogel sensitive to external load: (stress free) MNP network has fractal branches not yet wide open for EDL (electric double layer) build-up; (Compression) vertical pressure separates fractal branches apart ($d_c > d$ and $d_c' > d'$), promoting charge trapping (increased capacitance due to EDL insertion); and (shear) lateral pressure squeezes branches together ($d_s < d$ and $d_s' < d'$), closing gaps for EDL (reduced capacitance). **(b)** Array of MNP in the hydrogel can be used to map out contour of shaped loads of 8.0 kPa, from left to right for the shape of an 'O', an 'L' and a small dot. Capacitance increases (green; arrow up) or decreases (red; arrow down) due to location of the loads. **(c)** Angle-dependent sensing of airflows (red dot) suggests shear-induced pore closing in MNP for capacitance reduction. Inset photo illustrates the airflow angle ($\Theta$).

deformation is induced by acoustic waves[58]. These studies exploit one of the key strengths of the hydrogels, excellent acoustic impedance match to water. To verify this, we sealed the MNP–hydrogel device with a plastic wrap, soaked it in water and directed acoustic waves perpendicular towards the device. Figure 4a (for $10^{-2}$ mM salt concentration) and Figure 4b (for 100 mM salt concentration) illustrate how efficient the acoustic wave is at modulating the capacitance. Wave amplitudes of only 12 Pa produce robust signals with amplitude 23 mV. Clearly, a larger sound pressure gives rise to an increased signal (voltage), with the gel of higher ion concentration more sensitive to low pressures (Fig. 4b). As the measured signal is

from an oscilloscope and is due to a current flow through the external resistor ($R = 100$ k$\Omega$; Fig. 1c), we can further convert this voltage signal into capacitance change ($\Delta C$; Supplementary Note 2; Supplementary Fig. 4). Figure 4c shows a plot of the absolute capacitance change recorded as a function of sound pressure amplitude from 4 to 70 Pa (Supplementary Note 3; Supplementary Fig. 5). A careful look at these data suggests that the performance of this hydrogel microphone roughly falling into two regimes, with a higher sensitivity regime above a sound pressure of 30 Pa, but lower sensitivity below this value. Even in the lower sensitivity regime, the hydrogel device responds with orders of magnitude larger absolute capacitance change (nF) than

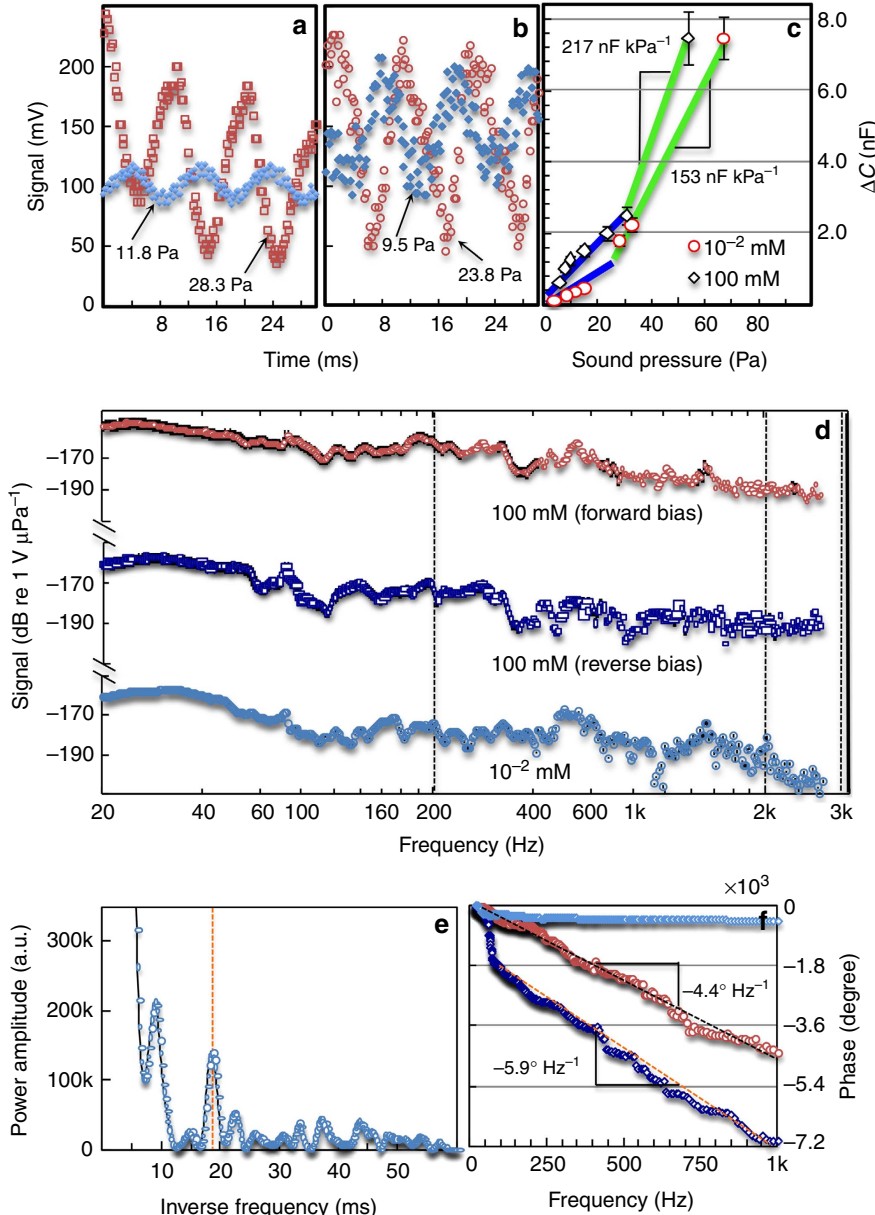

**Figure 4 | Response of hydrogel microphone to underwater acoustic waves.** Hydrogel microphone with (**a**) a low ($10^{-2}$ mM) or (**b**) high (100 mM) ion concentration responds to sound waves with different voltage output. (**c**) Capacitance change in microphone under sound pressures from 4 to 70 Pa, where gel membrane of 100 mM ions (unfilled squares) gives a higher sensitivity (217 nF kPa$^{-1}$; device area of 9 mm$^2$) than that of $10^{-2}$ mM (unfilled circles). (**d**) Frequency response of the hydrogel microphone from 20 Hz to 3 kHz, where ion concentration and bias direction affect the performance of the hydrogel microphone. Error bars (95% confidence) represent the variation of measurements due to sound interference inside the water tank. (**e**) Fourier transform on the frequency response further reveals a periodicity that is supposed to be caused by some form of wave propagating and interference through the thickness of the hydrogel. The wave velocity can be calculated from the periodicity (55 Hz) extracted in **e** and the thickness of the hydrogel (1 mm) to be 0.055 m s$^{-1}$. This is further verified by the phase delay measurement shown in **f**, where a linearly increasing negative phase shift can be observed in the hydrogel devices (red—$10^{-2}$ mM; dark blue—100 mM) but not the commercial hydrophone (light blue). This phase shift indicates a time delay of 15–20 ms, which suggest a wave velocity of 0.05–0.067 m s$^{-1}$ (using 1 mm hydrogel thickness). The nature of this slowly propagating wave in the hydrogel is suspected to be ion concentration wave.

conventional dielectric devices (pF)[19–22,46,47], primarily owing to the intrinsically large value of EDL capacitance and the efficient acoustic coupling of the gel membrane to water. For example, the base value of the MNP–hydrogel device at a low ion concentration ($10^{-2}$ mM) at 100 Hz is ~50 nF (Supplementary Fig. 6), Fig. 4c gives us a relative capacitance change ($\Delta C/C_0 \sim 14\%$ at 67 Pa) that appears comparable to the same device operated in air at 1 kHz of measurement frequency ($\Delta C/C_0 = 10\%$ at 1.0 kPa, Supplementary Fig. 3). An even more

revealing comparison is that the device sensitivity in water (217 nF kPa$^{-1}$) is over 2,000 times larger than in air (0.1 nF kPa$^{-1}$). Overall, this example confirms a high acoustic coupling efficiency of the gel membrane to underwater sound.

Consider the fact that the MNP network is surrounded by a hydrogel network that is relatively dense (3.0 M monomer concentration)[59], exposed to a high-frequency pressure wave, this dense network may not be able to keep up with the deformation rate of metal particles. As such, we can regard the

deformation of the MNP network and the hydrogel network as vibrations from a spring and a damper, respectively, where device response over a broad frequency will inform us general roles of these two structural units. Figure 4d showed response of this hydrogel microphone (amplitude of signal as dB) to sound over a wide range of frequencies, as recorded by a network analyser. A relatively flat response of $-152$ dB from 20 to 600 Hz is observed for a device with a high ion concentration (100 mM), accompanied by the signal gradually approaching the noise level of $-195$ dB at 3 kHz. However, when the bias for this microphone is reversed (copper negative and ITO positive), the device performance drops, with a weaker response of $-160$ dB (20–600 Hz) and then a noise level of $-195$ dB even at 2 kHz. This latter measurement indicates a level of asymmetry or anisotropy from the microphone, where cations at the MNP side apparently are less favoured towards additional EDL build-up. Since this type of anisotropy can be also found in ion-selective membranes[60] and can be interpreted as increased internal resistance for the motion of cations, energy conversion efficiency from a mechanical vibration (sound wave) to an electrical one (capacitance) therefore must be reduced. For low ion concentration ones ($10^{-2}$ mM), the device starts rather similarly with a flat response of $-165$ dB (20–600 Hz) then drops to a noise level of $-200$ dB at 3 kHz. Because the average salt concentration in ocean is $\sim 600$ mM, the hydrogel microphone containing this much of salt is slightly better than the one with 100 mM salt (Supplementary Fig. 7). A simple vibration model with a mass, a spring and a damper suggests both the MNP and hydrogel network structure play important roles in governing the response of the microphone at higher frequencies (Supplementary Note 4; Supplementary Fig. 8).

In addition to the general trend of decreasing response towards higher frequencies, small fluctuations can also be observed. If linear frequency axis is used, the fluctuation appears to be periodic. Indeed, a Fourier transform (Fig. 4e) of the frequency response curve shows a prominent peak at $\sim 18$ ms corresponding to $\sim 55$ Hz periodicity (peaks to the left of this one represents d.c. or slowly varying components that have been discussed previously). We attribute this to interference of waves residing in the hydrogel, similar to optical Fabry–Pérot interference. The nature of these waves is however not acoustic due to the small wave velocity of $\sim 0.055$ m s$^{-1}$ (using Fabry–Pérot interference equation, wave velocity is the hydrogel thickness of 1 mm times the frequency periodicity of 55 Hz). The existence of this slowly propagating wave is further confirmed by phase curves of the hydrogel microphone in Fig. 4f (note a linear scale of frequency is used) with near-constant negative slopes, indicating a time delay[61]. From the magnitude of the slopes, time delays of 15–20 ms are extracted, corresponding to a wave velocity of 0.05–0.067 m s$^{-1}$. This time delay, again, is not caused by the sound propagation time which is estimated to be $<0.1$ ms, as can be seen in the hydrophone measurement that has nearly zero slope (a sudden phase change near $\sim 50$–100 Hz is attributed to passing the resonating frequency of hydrophone and loudspeaker, respectively[61]). We suspect an ion concentration wave has caused such a nontrivial delay and the previous fluctuations in response curve. Possibly, ion concentrations are quickly modulated by acoustic waves that change the surface area of the silver nanoparticle electrode, yet it does not immediately induce electrical current until the perturbation of the ion concentration reaches the counter electrode, where the resistor is connected to. A crude model of such is formulated and can be found in Supplementary Note 5. We note the existence of this ion concentration wave in a solid or gel has never been documented before despite its well-known counterpart (ion acoustic wave) in plasma[62,63]; therefore, this new phenomenon deserves more

efforts in the future, especially on their dependence on device structure, electrical bias, as well as properties of the gel membrane.

## Discussion

The deformable dendritic network of MNPs is a critical component of the hydrogel capacitor, enabling it to efficiently detect airflow, touch and underwater acoustic waves. This sponge-like superstructure grows from the photocathode by an electrochemical reduction of dispersed silver ions inside a hydrogel matrix. Under external load, these metallic superstructures bear two roles in capacitance tuning, one through deformation to change the electrode area for EDL build-up or removal and the other through the applied bias and the deformation of MNP network to modulate the ion concentration in gel body. In comparison with energy generators of piezoelectric mechanism[27], even though the device operates as an energy storage device, it provides a remarkably large responsivity of 24 $\mu$C N$^{-1}$ at a bias of 1.0 V, whereas crystals of Pb(Mg$_{\frac{1}{3}}$Nb$_{\frac{2}{3}}$)O$_3$-PbTiO$_3$ have responsivities of only 2.8 nC N$^{-1}$ (refs 27,28). While soft and flexible materials play an increasingly large role in the current era of smart and portable electronics[21,64,65], this work illustrates the opportunities afforded by ion-rich EDLs in hydrogels. As this mechanism is further optimized and exploited, we expect that the excellent acoustic impedance match to water, the ease in processing and manufacturing, and greatly increased sensitivities afforded by the EDL modulation approach will lead to numerous uses in human–computer interfaces as well. Quite surprisingly, the microphones also exhibit a highly intriguing ion wave resonance. Because the hydrogel matrix functions as a transparent skeleton to underwater acoustic waves and as an ion reservoir, transient modulation of EDLs creates a packet of ionic waves, moving from the MNP-planted side to the MNP-free side. As such, response of this hydrogel capacitor is not only sensitive to internal ion concentrations, but also differs from traditional dielectric- or piezoelectric-based devices by delivering an ionic wave-superimposed response every 55 Hz, a phase lag of 15–20 ms, as well as an unmatched performance at low frequencies.

## Methods

**Synthesis of metal ion-loaded hydrogel membrane.** We chose Ag$^+$ as the source of metal ions and there is no major difference in synthesizing a pristine hydrogel versus Ag$^+$-loaded one. Generally, 10 ml aqueous solution of acrylamide (monomer; 3.0 M, 2.133 g), methylene bisacrylamide (crosslinker; 10.0 mol%, 0.456 g), ammonium persulfate (initiator; 0.05 mol%, 0.003 g) and AgNO$_3$ (30.0 mM, 0.051 g) were mixed in a 50 ml plastic beaker. Then, the solution was pipetted to fill a small volume between two parallel glass sheets that were separated by a silicone spacer of 1.0 mm in thickness. Ag$^+$-loaded PAAm hydrogel membrane was peeled from the substrate and cut into desired shape and size after 1 h of gelation at room temperature.

**Converting metal ions into network of MNPs.** To control the growth site of MNP in hydrogels, the Ag$^+$-loaded PAAm membrane was first placed on top of an amorphous silicon substrate (a-Si; 3.4 × 2.3 cm$^2$; Solar-powered Polyresin Rock Garden Lights; Greenbrier International Inc.) that was previously soaked in a dilute solution of hydrochloride (2.0 M, 50 ml) to etch away the aluminum cover and later rinsed with copious DI water (100 ml). Then, an ITO (Sigma Aldrich)-coated glass slide was covered on top of this hydrogel membrane, followed by exposing the a-Si surface with a projected light from an optical microscope (Meiji, Japan) and simultaneously biasing the ITO as anode and a-Si as cathode under a voltage of 3.0 V for 5 s. This photoconduction from a-Si effectively triggered the formation of a dark coloured network of Ag nanoparticles inside the hydrogel membrane.

**Copper coating on hydrogel.** MNP–hydrogel was first immersed in DI water for 48 h to remove excess Ag$^+$ inside the host membrane (DI water replaced every 12 h). Then, the ion-free membrane was transferred to a plating solution (10 ml) consisting of CuSO$_4 \cdot 5H_2O$ (80 mM, 0.18 g), ethylenediaminetetraacetic acid (165 mM, 0.48 g), K$_4$[Fe(CN)$_6$] (150 $\mu$M, 0.6 mg), NaOH (adjust pH to 12.8, 0.04 g)

and formaldehyde (300 mM, 225 µl) for 60 min at room temperature. Once a smooth layer of copper is coated atop the MNP–hydrogel, it was rinsed with DI water and kept wet until a later use.

**Fabrication of static pressure sensor.** To map the shape of a static load or pressure, a $4 \times 4$ sensor array was fabricated first by growth of four $3 \times 27$ mm stripes of Ag nanoparticles on one side of the PAAm hydrogel membrane and then another four stripes orthogonally on the other side, followed by electroless copper plating (Supplementary Fig. 9). Each copper stripe was then connected with a thin copper wire (12 G) through a silver paste (Electron Microscopy Sciences), followed by further coating with epoxy glue (Loctite Quick Set Epoxy) to secure a firm contact for the later capacitance measurement. A $3 \times 3$ mm square of Ag nanoparticles patch was fabricated in the shallow surface of hydrogel, followed by electroless copper plating to study the capacitance change under different salt concentrations and different static loads.

**Static pressure measurement.** A weight of 5.0 g (square shape with an area of 9 mm$^2$) was placed on top of the hydrogel device (with MNP or without), with a pressure calculated as 5.4 kPa, to determine the capacitance change in the device under various salt concentrations. For pressure mapping, a static load was applied by placing a weight of 66, 59 or 3.0 g on top of the copper-coated MNP–hydrogel (device size of 81 ('O' shape), 72 ('L' shape) and 4 mm$^2$ (small dot)), corresponding to a pressure of 8.0 kPa. To study the static load sensitivity, a series of weights ranging from 0.9 to 9.0 g were used, with pressures calculated from 1.0 to 10.0 kPa. For the detection of air movement, a piece of copper-coated MNP–hydrogel with $3 \times 3$ mm area of Ag nanoparticles was used as the sensing device (ITO as another electrode). The airflow was generated by gently squeezing a rubber blower fixed on a ring stand, where the gas release duration is controlled, with the pressure being read by placing the syringe close to a digital balance. The angle of blowing is adjusted to ensure the tip of the blower always pointing to the centre of the Cu electrode. The distance between the tip of rubber blower and the surface of hydrogel was fixed at 10 mm, and the pressure generated by gripping was controlled at 1.0 kPa.

**Preparation of salt-loaded hydrogel devices.** Hydrogel membrane (PAAm only or copper-coated one) was immersed in a plastic beaker that contains an aqueous solution of NaCl with a concentration of $10^{-2}$, 1, 10, 100 or 1,000 mM for 4 h (note: ion concentration inside the gel can be much less, dependent on the density and chemical structure of the gel[66]). Then, the copper-coated MNP–hydrogel was wiped dry and wired with an ITO plate ($15 \times 15$ mm$^2$) for capacitance measurement. In the case of MNP-free one, a piece of aluminum foil ($3 \times 3$ mm$^2$) was used as a counter electrode to pair up with ITO plate for capacitance probing.

**Scanning electron and confocal laser scanning microscopy.** The Ag-implanted PAAm hydrogel is sectioned into $\sim 60$ µm thick slices (Vibratome 1000) to expose the cross-sectional profile and subsequently loaded into a super critical point dryer (Samdri 780A) to remove the water content without causing significant structural change to the hydrogel. The dried slice is then glued on a metal plate and examined directly with a Hitachi S-4700 FE-SEM (field emission scanning electron microscope) under a voltage of 10–20 kV. For confocal microscopy, a bulk piece of Ag-implanted PAAm gel is placed on top of a glass side, scanned with a $\times 60$ water immersion lens and imaged using a Nikon A1 confocal laser scanning microscope system on a Nikon Eclipse 90i using the 561.4 nm excitation laser line (Nikon Instruments Inc., Melville, NY).

**Capacitance measurement.** Capacitance of the hydrogel depends heavily on frequency and voltage applied (Supplementary Fig. 6). We chose 1 kHz and 20 mV (parallel mode) for the capacitance measurement due to its relative insensitivity to measurement frequencies and minimal interference to electrolytes inside the hydrogel membrane. All the capacitances were measured by using Hewlett Packard (4263B) LCR meter, and experimental data were exported to a LabVIEW program.

**Underwater acoustic wave detection.** A plastic wrap is used to insulate the hydrogel device from its aqueous environment, therefore no current shorting or ion infiltration between the device and its liquid background occurs. Then, the device was placed inside a home-made metal mesh cage and immersed in water, by connecting the leads of the device with an external resistor (100 kΩ) via a home-designed circuit (Fig. 1c) to convert capacitance change into voltage output. A computer program-controlled loudspeaker (Dayton Audio, DAEX25VT-4 Vented 25 mm Exciter 20 W 4 Ohm) driven by an amplifier (Lepai, LP-2020A + Tripath TA2020 Class-T Hi-Fi Audio Amplifier) served as the acoustic wave generator, from which both frequencies and amplitudes are adjusted. An oscilloscope (Rigol DS1102E) was used to record the voltage output on the external resistor (100 kΩ). For frequency sweeping and phase lag measurements, a network analyser (Hewlett Packard 3577A) was used as both an acoustic wave generator and a signal receiver, with the experimental data collected by a customized LabVIEW program.

**Calibration of underwater sound pressure with hydrophone.** To determine the local sound pressure applied on the hydrogel microphone, a commercial hydrophone (SQ 26 Cetacean Research Technology, Seattle, WA) was used to replace the hydrogel device at the same location under identical acoustic impact conditions (settings controlled by the sound card on computer and the amplification ratio). Once voltage output from the hydrophone is recorded by an oscilloscope, sound pressure with unit of Pa was calculated using peak-to-peak voltage and sensitivity map of the hydrophone (Supplementary Note 3; Supplementary Fig. 5).

**Spectrogram and audio recording.** Audio input in the form of a sine sweep with constant amplitude over frequencies from 20 Hz to 3 kHz is applied to the MNP–hydrogel microphone. The signal recorded was analysed for the frequency content by Fourier transform analysis. A short speech (Supplementary Movie) was also recorded using this MNP–hydrogel microphone, without using any low-noise pre-amp or installing electromagnetic shielding around the apparatus.

**Data availability.** The authors declare that the data supporting the findings of this study are available within the article and its Supplementary Information files.

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

## Acknowledgements

L.T. gratefully acknowledges the financial support from the National Science Foundation (grant number: CMMI 1098652 and IIA 1338988) and J.A. Woollam Foundation. Y.M.C. acknowledges the financial support from the Shaanxi Province (International Science & Technology Cooperation Program, 2013KW14-02; Fundamental Research Funds for the Central Universities, the Program for the Key Science and Technology Innovative Team, 2013KCT-05). S.L. thanks an educational grant from the China Scholarship Council. We also thank Mr. Shichao Li (Dalian Institute of Technology) for verifying some of the experiments and collecting extra data during the manuscript revision process.

## Author contributions

L.T., Q.Z. and Y.M.C. conceptualized the work. Y.G. carried out most of the device development and experimental work. J.S. helped with apparatus design and data collection through LabView programming. S.L. helped with metal deposition using digital exposures. C.E. and Y.Z. assisted in confocal microscopy imaging. Q.Z. helped with many experimental set-ups, underwater acoustic measurements, as well as dynamic modelling on ion concentration waves. L.T. wrote the first draft of the manuscript and S.D. thoroughly revised the final version. All authors have given approval to the final version of the manuscript.

## Additional information

**Competing financial interests:** The authors declare no competing financial interest.

