## [Peer Review File · Nature Communications]

Reviewers' comments:

Reviewer #1 (Remarks to the Author):

In their manuscript "Hydrogel Microphone: Deformable Network of Metal Nanoparticles Enables Stealthy Underwater Listening with Extreme Sensitivity", Yang Gao, Jingfeng Song, Shumin Li, Christian Elowsky, You Zhou, Stephen Ducharme, Yongmei Chen, Qin Zhou and Li Tan describe a new approach for measuring static and dynamic mechanical loads at very high sensitivity. This new approach is based on mechanical to electrical coupling via variable electric double layer capacitors. The variability in the electrical double layer capacitors stems from mechanically induced changes in the microstructure of deformable networks of metal nanoparticles (MNP) inside of ionically conductive hydrogels. The MNP network is composed of fractal branch structures, where mechanical stimuli change the distance between single branches. For specific locations, the room between two branches can become too small to allow for the formation of electrical double layers. This effect can be translated into a variable area available for the formation of electrical double layer capacitors. Therefore, changes in mechanical pressure become related to changes in overall capacitance of the device.

The authors demonstrate this new mechanism for static and for dynamic mechanical loads and place particular emphasis on potential applications in the area of underwater sound detection with high sensitivity for the low frequency regime.

The manuscript presents a new and very creative approach with a range of potential applications, likely beyond the area of underwater sound detection. This paper clearly has the potential to inspire follow up work into diverse new directions, and thus seems suitable for the readership of Nature Communications. Nevertheless, in the current form, there are a number of issues that need to be addressed in detail before I would recommend this paper for publication:

A) While the majority of experiments seems to be designed and analyzed carefully and correctly, the quality of presentation throughout the paper is not up to the standards required for this journal.

- The quality of the language throughout this manuscript makes it hard and unnecessarily cumbersome to understand central scientific points. In particular the last two paragraphs of the introduction are full of sentences that lack the necessary clarity to unambiguously convey ideas. Additionally, the section "Fabrication of MNP-Hydrogels" leaves a lot of things unclear, mainly due to awkward formulations, and unstructured jumps between different thoughts. A representative example of the type of sentences that need to be updated: "As silver reduction process critically calls a fast diffusion of metal ions, it is likely that the fractal network of MNP formed inside those water-rich gaps".

- Table 1: It is not clear whether statistics was used here or not. Did the authors measure different samples, how many, and how large are the observed experimental errors?

- Figure 1 C: This subfigure is not clear enough and needs to be improved. How is the second electrode connected to the hydrogel (this becomes clear after reading the text in detail, but from the figure this is left completely unclear)? What is the black rectangle around the hydrogel sample, and what does the purple shading indicate?

- Figure 2 D: How were the error bars obtained? How many samples were used? Did the authors use

different samples to get error bars or did they use the same samples with multiple tests performed on them?

- Figure 4 C: The use of statistics and error bars would enhance the significance of these results.
- Figure S1: Could the authors provide a higher resolution image here? A lot of detail is blurred out in the current version.
- Figures S3, C and D; Figure S4, A and B: Graphs are not labeled clearly, and superimposing of multiple signals makes it impossible to decipher information on these graphs.

B) Capacitance Measurements:

In the methods sections, the authors state that all capacitances are measured with a LCR meter at 1 kHz and at 20 mV. Given the fact that measured capacitance is a key element in the scientific analysis of the experimental results this limitation to a specific frequency and voltage could be a qualitative and quantitative problem. Literature on electric double layer capacitors indicates, that the capacitance of electric double layers is a function of voltage used to measure the capacitance, as well as of frequency of the measurement signal. Also, the common method to determine capacitance of supercapacitors (based on electrical double layer capacitors) is cyclic voltammetry; exploration of this method could potentially provide useful further insights into the mechanism of variable capacitance analyzed in this paper. Moreover, for measurements with LCR meters, it is important to include information about whether a parallel or series mode was used in the settings -- electrical double layer capacitors are known to give different results depending on which mode is used.

C) Suggested improvements and requests for further discussion:

- Figure 1 D: Why is this signal so different from a sine wave? Is this deviation caused by a superimposed noise signal?

Figure 2 D: Purple curve: Why is there a peak at 100 mM and a subsequent drop in the capacitance change? Some info should be included in the caption (or at least a hint towards a discussion in the text).

- Figure 3 C (inset photo). How did the authors control and measure the pressure on the bulb syringe (was this controlled by hand and estimated)? More generally, the use of air pressure to analyze device behavior is not motivated in the paper.
- DOI: 10.1021/ja7106178; "Relation between the Ion Size and Pore Size for an Electric Double-Layer Capacitor"; this paper could be a useful reference to discuss the suggested mechanism for capacitance change in the electrical double layer capacitors.
- The description of the static pressure tests could be clarified with a schematic (maybe include that in the supplementary section).
- Throughout the paper, the authors are using different ion concentrations. It would be useful to include typical ion concentrations in the ocean, as this device is intended for underwater audio detection.
- The main message of this paper is focused on a new idea to fabricate a high-end audio device. This

message is not supported convincingly enough with acoustic experiments and demonstrations. An additional experiment could be the creation of a spectrogram (see ref 28 of the paper): audio input in the form of a sine sweep with constant amplitude from low frequency to high frequency. The signal recorded with the new microphone should be analyzed for frequency content by Fourier transform analysis. This would give important additional info about the quality of the recording, such as: Do the complex micro-architecture of the microphone and the suggested physical mechanism introduce any kind of higher harmonics, increased sensitivity to specific frequencies, resonances, etc.?

- The central message of the paper (extremely sensitive hydrogel microphone) would be strongly supported by providing an audio sample of recordings performed with the hydrogel microphone. How about playing a piece of music or part of a speech over the loudspeaker and providing the recorded sound file in the supplementary materials? Of course, the cut-off at higher frequencies could be a problem, but 3 kHz should be high enough to generate a meaningful result. In any case, this type of data would give readers a much more intuitive feel about the performance of this device.

Reviewer #2 (Remarks to the Author):

The authors have reported a capacitive sensor by using fractal-structured metal nanoparticles on hydrogel matrices and showed the ability of signal sensing under mechanical stimuli in three cases; airflow, static loads and underwater acoustic waves. They utilized the feature of large capacitance from hydrogels enabled by electrical double layer and invented the method for fabricating deformable Ag fractal networks in intended patterns. Their suggestion on the application of their devices as hydrogel microphones underwater is quite interesting but there are some unclear factors I would like to ask and minor points which should be revised. Please consult the followings:

1. In page 12, the absolute sensitivity of the MNP-hydrogel device seems better than conventional dielectric capacitors but the relative sensitivity, which would be more appropriate to be an evaluation criterion of sensing, does not. This makes the description of "extreme sensitivity" in the title unsuitable.
2. The overall accounts on Figure 4D are vague. There should be more specific descriptions, especially on the high frequency regions. In addition, when the frequency follows a cyclic pattern, does the MNP-device maintain identical detection behaviors? Is there any chance to show a hysteresis?
3. In Figure 3A, the illustration and its caption can be misunderstood - Since the distance of EDL, which is in atomic size, does not affected by external stimuli (Conway B.E., et al., Modern Aspects of Electrochemistry v.31, Kluwer, 1999), there should be no variation in gaps of EDL. The explanation on capacitance variations by external stimuli needs to be revised more clearly.
4. Is there any relationship between the sensitivity of the device and mechanical properties of hydrogel matrices?
5. There is no information on input signals for detecting underwater sound in Figure 1D. Input signals are expected to be added on the graph for comparison with the output.
6. In page 12, authors have mentioned an additional factor on the variation of EDL capacitance by ion concentration modulation (excluding mechanical stimuli) - the fact that MNPs are electrically biased. Is there any complementary explanation on the effect of electrical bias on the MNP-hydrogel device?
7. How did you measure the sound pressure in Figure 4C?
8. The word "fractal structure" is confusing. Please consider an alternative.
9. At the 6th line in page 4, I cannot catch up with the meaning of "figure 8."
10. Schematics in Figure 3B and 3C (especially the inset photo) are not effective to deliver the information.
11. In the manuscript for Figure 2D, the explanation is unclear and notations in Figure 2D are not eye-

catching. There should be some modifications.
12. It would be better to adding a legend in Figure 4C.

Reviewer #3 (Remarks to the Author):

Evaluation of Hydrogel Microphone: Deformable Network of Metal Nanoparticles Enables Stealthy Underwater Listening with Extreme Sensitivity
Authors Yang Gao et al

The authors describe a microphone that combines hydrogel with metal particles, where sound on system supposedly causes capacitance changes. The claim is that the good impedance match of water to hydrogel makes the microphone stealthy (this term even appears in title of paper).

I find that while the work has some interesting features, it is highly overblown and not suitable for publication in this journal.

The authors assert that this microphone is the only obstacle to make submarines totally undetectable. This is laughable! Either the authors do not understand how acoustic or EM cloaking really works, and they do not appreciate the complexities of sound suppression of submarine propulsion, or they are intentionally misleading reader.

Second, although the configuration presented is said to be capacitive, there are strong similarities to conventional carbon pack microphone, where small carbon particles are loosely packed (often with no direct acoustic chamber) such that acoustic wave impinging causes particles to change connectivity (and capacitance!). That microphone invented in the 1870s was the basis of the telephone industry for over 100 years! Why no reference to this?

The authors compare impedance mismatch of hard piezo material to hydrogel. But there are many non-piezo microphone systems around. Even humans can hear really well underwater (in fact better than in air, that is why they lose stereo direction information). And they don't use piezos either.

This device seems to be sensitive to ion infiltration. So fluctuations in salinity (example) give different and likely uncontrolled response.

This work is best for a journal other than Nature Communications.

REVIEWERS' COMMENTS:

Reviewer #1 (Remarks to the Author):

The authors have addressed the concerns of the all three reviewers in a reasonable manner. With the constructive criticism the paper has become a more convincing piece of work. I continue to strongly suggest a native English speaker assist authors in making a more fluid and coherent text that does not hinder the reader's ability to understand the science. I recommend publication of this article after the additional points below have been addressed:

1) Although some areas of the paper have improved, quality of writing is still below the level I would expect for this journal. Sentences are over complicated and it is sometimes difficult to follow the writing. Maybe have a native speaker revise the article.

2) Use of statistics in Table 1 (and throughout the manuscript): Different samples, how many, and how large were the experimental errors?

For Table 1, according to authors "we used two ways to check repetitivity or variability of our hydrogel samples" one by using multiple samples (3 or more) and with the same sample but 3 times of data recording.

Here, I don't think the following explanation is clear enough: "rather constant value in relative standard deviation (standard deviation over average)". Also, where do the values for standard deviation come from? To simplify the explanation of the statistics used, could the authors say "3 or more samples were tested multiple times"?

Most importantly, the exact statistical analyses that were used throughout the manuscript should be clearly conveyed to the reader. If three samples were used the reader should have access to this information.

3) Figures S5 c) and d) show voltage amplitude from hydrogel for different ion concentrations. The voltage scales are different here. If the vertical scales for each plot from the 10^{-2} and 100 mM were the same (as input signal increases and between different ion concentrations) the effects of increasing input and ion concentration would be clearer.

4) The signal shown in figure 1 D is different from a sine wave. The authors explain in their rebuttal letter that the sound transfers from the amplifier and loudspeaker to the water tank and thereby becomes non-linear. It may be helpful to add this information to the paper or supplemental -- it is possible other readers have this same question.

5) The authors explain in their rebuttal letter that airflow from the bulb syringe is controlled by timing how long it takes until the bulb is evacuated. This information should be added to the paper in the methods section.

Reviewer #2 (Remarks to the Author):

My opinions on the publication of this paper in Nature Communications have not considerably changed. If accepted, many who read this will probably view an interesting capacitive sensor whose future application lies on salty underwater areas. Demonstration of hydrogel microphones with silver networks whose capacitance values vary with stimuli is attractive itself, and the revised version,

including capacitance measurements in Figure S2 and pressure test in Figure S4 in supplementary information, provides more details on what they have done with clarity. There are still many theoretical issues including the effects of coupling of waves and ion concentration waves on the MNP-hydrogel sensor, but they are not too late to be dealt in further studies. I expect more delight studies on MNP-hydrogel microphone in terms of theories and the best use of this device. I am in favor of publication.

Reviewer #1 (Remarks to the Author):

In their manuscript "Hydrogel Microphone: Deformable Network of Metal Nanoparticles Enables Stealthy Underwater Listening with Extreme Sensitivity", Yang Gao, Jingfeng Song, Shumin Li, Christian Elowsky, You Zhou, Stephen Ducharme, Yongmei Chen, Qin Zhou and Li Tan describe a new approach for measuring static and dynamic mechanical loads at very high sensitivity. This new approach is based on mechanical to electrical coupling via variable electric double layer capacitors. The variability in the electrical double layer capacitors stems from mechanically induced changes in the microstructure of deformable networks of metal nanoparticles (MNP) inside of ionically conductive hydrogels. The MNP network is composed of fractal branch structures, where mechanical stimuli change the distance between single branches. For specific locations, the room between two branches can become too small to allow for the formation of electrical double layers. This effect can be translated into a variable area available for the formation of electrical double layer capacitors. Therefore, changes in mechanical pressure become related to changes in overall capacitance of the device.

The authors demonstrate this new mechanism for static and for dynamic mechanical loads and place particular emphasis on potential applications in the area of underwater sound detection with high sensitivity for the low frequency regime.

The manuscript presents a new and very creative approach with a range of potential applications, likely beyond the area of underwater sound detection. This paper clearly has the potential to inspire follow up work into diverse new directions, and thus seems suitable for the readership of Nature Communications. Nevertheless, in the current form, there are a number of issues that need to be addressed in detail before I would recommend this paper for publication:

A) While the majority of experiments seems to be designed and analyzed carefully and correctly, the quality of presentation throughout the paper is not up to the standards required for this journal.

1. The quality of the language throughout this manuscript makes it hard and unnecessarily cumbersome to understand central scientific points. In particular the last two paragraphs of the introduction are full of sentences that lack the necessary clarity to unambiguously convey ideas. Additionally, the section "Fabrication of MNP-Hydrogels" leaves a lot of things unclear, mainly due to awkward formulations, and unstructured jumps between different thoughts. A representative example of the type of sentences that need to be updated: "As silver reduction process critically calls a fast diffusion of metal ions, it is likely that the fractal network of MNP formed inside those water-rich gaps".

To add clarity to the claims, we simplified some sentences but added details to a few other ones in the last two paragraphs and the first section of the Results and Discussion:

- (a) Changed "*Once this cavity-free device...*" on Page 4 to "*Once this MNP-hydrogel device...*"
- (b) Shortened "*the MNP-hydrogel device...*" on Page 4 to "*it*"

- (c) Changed *“device response can be extended to acoustic waves of kilohertz...”* to *“the device can respond to acoustic waves at kilohertz...”*
- (d) Changed *“a typical “figure 8” pattern with the maximum at normal incidence”* on Page 4 to *“having a maximum value when sound incidence is perpendicular to the surface of the hydrogel microphone but a much weaker one when sound is parallel to the microphone surface or a typical “figure 8”-like pattern”*
- (e) Added one sentence at the end of the Introduction on Page 4: *“Unlike David Hughes’ early version of acoustic transmitter (carbon microphone) that works by compacting loosely connected carbon particles between two metallic plates with a change in resistance, our device is completely void or cavity-free, implying zero-reflection toward SONAR scanning.”*
- (f) Changed *“Making a fractal network of MNP...”* on Page 5 to *“Making a continuous network of MNP...”* and *“...make the MNP”* to *“...make such a superstructure”*
- (g) Changed *“...a piece of hydrogel is first soaked inside a dilute solution of silver nitrate (AgNO₃)...”* on Page 5 to *“silver nitrate (AgNO₃) is preloaded into a piece of hydrogel during the hydrogel synthesis or later added by soaking hydrogel inside a dilute solution of the silver ions...”*
- (h) Shortened *“...the anode (positive bias) must be an active metal or metal oxide like aluminum or ITO in order to achieve a quick implanting (5 sec) under a relatively high bias (3.0 V). Sacrifice of these solids into metal ions prevents water electrolysis or decomposition inside the hydrogel matrix and hence, no oxygen bubbles are produced to penetrate or tear the entire gel membrane”* on Page 5 to *“...the anode (positive bias) must be expendable like aluminum or ITO in order to prevent water electrolysis or decomposition inside the hydrogel”*
- (i) Changed *“a-Si can be used as the electrode”* on Page 5 to *“a-Si can be used as the contact electrode (Steps 2 & 3)”*
- (j) Shortened *“In aforementioned steps, photosensitive a-Si is used as the cathode to allow digital exposure as one efficient way to control the growth sites for MNP. Since small patterns are generated through a digital projector that is connected with an optical microscope, the pattern shape and size (7.6 μm) can be easily modified”* on Page 6 to *“As a-Si is photosensitive, we can further use digital exposure as one efficient way to control the growth sites for MNP. Here, small patterns are generated through a digital projector that is connected with an optical microscope”*
- (k) Changed *“Alternatively, we can interpret the silver growth by looking at the structure defects inside the gel membrane”* on Page 7 to *“From materials physics point of view, we can interpret the silver growth being occurred at the structural defect sites of the gel membrane”*
- (l) Changed *“As silver reduction process critically calls a fast diffusion of metal ions, it is likely that the fractal network...”* on Page 7 to *“Water-rich gaps however can be regarded as boundaries between neighboring polymer blobs or simply as structural defect sites. As most water molecules inside these gaps are free to move, a quick silver reduction triggers the formation of a continuous network of silver NPs”*
- (m) Changed *“electrode”* to *“contact electrode”*, *“fractal shape”* to *“spongy network of metal NPs”* in the last sentence of the section of *“Fabrication of MNP-Hydrogel”* on Page 7.

2. Table 1: It is not clear whether statistics was used here or not. Did the authors measure different samples, how many, and how large are the observed experimental errors?

Yes, statistics are used for Table 1 and Figure 2D. We used two ways to check repetitivity or variability of our hydrogel samples, one by using multiple samples (at least 3) at the same salt concentration level and the other through the same sample, but with more than 3 times of data recording. Once the salt concentration is fixed and the sample is maintained well (no deformation due to drying), we found consistent measurement in absolute capacitance change (ΔC) or relative capacitance change ($\Delta C/C_0$, %), with some fluctuation in standard deviations. For instance, when salt concentration is fixed for MNP-hydrogel at 10^{-2} mM, multiple samples gave us an average ΔC of 0.73 ± 0.18 nF ($\Delta C/C_0$ is 23 ± 0.94) whereas multiple tests for the sample gave us 0.83 ± 0.04 nF ($\Delta C/C_0$ is 24 ± 1.2); when salt concentration is set at 100 mM, then respectively multiple samples have a ΔC of 14.0 ± 2.91 nF ($\Delta C/C_0$ is 8.0 ± 0.35) whereas multiple tests have the ΔC of 14.0 ± 0.72 nF ($\Delta C/C_0$ is 8.2 ± 0.42). By comparing samples tested at different salt conditions, we obtain a rather constant value in relative standard deviation (standard deviation over average). For ΔC , this value is 21-25% while for $\Delta C/C_0$ it is 3.5-4.5%. For MNP-free hydrogel samples, the relative standard deviation for ΔC is 13.0-13.5% while for $\Delta C/C_0$ it is 3.0-4.0%.

We added one sentence below Table 1 to illustrate the validity of the data inside the table: “*Note: Statistic measurements using multiple samples or the same sample with multiple tests are used. Relative standard deviation (standard deviation over average) of ΔC and $\Delta C/C_0$ for MNP-hydrogel is respectively 21-25% and 3.5-4.5%. For MNP-free hydrogel, relative standard deviation of ΔC and $\Delta C/C_0$ is respectively 13.0-13.5% and 3.0-4.0%.*”

3. Figure 1C: This subfigure is not clear enough and needs to be improved. How is the second electrode connected to the hydrogel (this becomes clear after reading the text in detail, but from the figure this is left completely unclear)? What is the black rectangle around the hydrogel sample, and what does the purple shading indicate?

We added a label next to the blue layer to indicate it as an indium-tin oxide (ITO) coated bottom electrode. We also added “water” inside the purple shading area to indicate the liquid medium. The black rectangle itself is the hydrogel sample. To avoid confusions, we changed the color to white (same color as the translucent sample in Figure 1A), plus a label of “hydrogel”.

4. Figure 2D: How were the error bars obtained? How many samples were used? Did the authors use different samples to get error bars or did they use the same samples with multiple tests performed on them?

We checked repetitivity of our hydrogel samples with multiple samples and for some of them, performed multiple tests to obtain an average value plus a standard deviation. Details please see answer to the Question #2.

5. Figure 4C: The use of statistics and error bars would enhance the significance of these results.

We now added error bars in Figure 4C. The statistics is done on the same sample under multiple tests.

6. Figure S1: Could the authors provide a higher resolution image here? A lot of detail is blurred out in the current version.

The current picture is the highest resolution we can possibly get. This limitation mainly comes from the technique used (confocal microscopy, where features below are covered up by features above). Therefore the main purpose of this supplemental figure is to show the 3D distribution of the silver nanoparticle network. To see the detailed structure of the network closely, please refer to the SEM picture (Figure 1) in the main text.

7. Figures S3, C and D; Figure S4, A and B: Graphs are not labeled clearly, and superimposing of multiple signals makes it impossible to decipher information on these graphs.

We separated those superimposing curves and re-plotted them as in Figures S5 and S6 respectively (two new figures have been inserted before the original Figures S3 and S4). In addition, we highlighted the change in signal input intensity (S5, acoustic source) and output intensity (S6, hydrogel microphone) next to those figures.

B) Capacitance Measurements:

8. In the methods sections, the authors state that all capacitances are measured with a LCR meter at 1 kHz and at 20 mV. Given the fact that measured capacitance is a key element in the scientific analysis of the experimental results this limitation to a specific frequency and voltage could be a qualitative and quantitative problem. Literature on electric double layer capacitors indicates, that the capacitance of electric double layers is a function of voltage used to measure the capacitance, as well as of frequency of the measurement signal. Also, the common method to determine capacitance of supercapacitors (based on electrical double layer capacitors) is cyclic voltammetry; exploration of this method could potentially provide useful further insights into the mechanism of variable capacitance analyzed in this paper. Moreover, for measurements with LCR meters, it is important to include information about whether a parallel or series mode was used in the settings -- electrical double layer capacitors are known to give different results depending on which mode is used.

Cyclic voltammetry is indeed the standard method. For sensing purposes we found 1 kHz at 20 mV (parallel mode) as the measurement platform gives us faster and more consistent results (see figures above). However, the hydrogel is an ionic conductor where the motions of the ions are much slower than electrons in solid. Conventional cyclic voltammetry (lower right corner of Figure) can capture the whole charge/discharge process and gives much larger value for the equilibrated capacitance. The “transient capacitance” measured at 1 kHz, on the other hand, gives us much faster response and more consistent pressure sensing results. Parallel mode is used for our measurements.

On Page 20, we added the following descriptions: *“Capacitance of hydrogel depends heavily on frequency and voltage applied (see Supplementary Information). We chose 1 kHz and 20 mV (parallel mode) for the capacitance measurement due to its faster response, relative insensitivity to measurement frequencies and minimal interference to electrolytes inside the hydrogel membrane.”* We also added above figure as Figure S2 in the Supplementary Information.

Meanwhile, on Page 14, we modified the last three sentences of the section on “Device Response to Underwater Acoustic Waves” to highlight the difference in frequency induced capacitance change vs. pressure induced changes: “For example, the base value of the MNP-hydrogel device at a low ion concentration (10^{-2} mM) @ 100 Hz is in the range of 50 nF (Figure S2), Figure 4C gives us a relative capacitance change ($\Delta C/C_0 \sim 14\%$ @ 67 Pa) that appears comparable to the same device operated in air @ 1 kHz of measurement frequency ($\Delta C/C_0 \sim 10\%$ @ 1.0 kPa, Figure S3). However, if we count the capacitance

Figure. (First row and lower left panel) capacitances of MNP-hydrogels with salt concentrations of 10^{-2} , 10, and 100 mM show a flat response only when the measurement frequency is above 1 kHz. (Lower right panel) cyclic voltammetry (10 mV/s scan) gives these samples a capacitance value of 243, 560, and 1293 μF respectively.

change per unit pressure or compare the absolute sensitivity in air (0.1 nF/kPa, Figure S3) versus that in water, three orders higher sensitivity (217 nF/kPa) is received, way beyond the frequency induced capacitance change in measurement (Figure S2). Overall, this example confirms a high acoustic coupling of the gel membrane to underwater sound”

C) Suggested improvements and requests for further discussion:

9. Figure 1 D: Why is this signal so different from a sine wave? Is this deviation caused by a superimposed noise signal?

When sound is transferred from our amplifier and loudspeaker to the water tank, the sound become nonlinear. We learned this by measuring the same 2 kHz input signal using a commercial hydrophone (SQ 26 Cetacean Research Technology, Seattle, WA), the response signal also showed zigzag-like curves. Details of this nonlinear behavior are better addressed in Question #15.

10. Figure 2 D: Purple curve: Why is there a peak at 100 mM and a subsequent drop in the capacitance change? Some info should be included in the caption (or at least a hint towards a discussion in the text).

We attributed this sudden drop to the EDL thickness dependence on ion concentrations. When ion concentration is low, thick EDL cannot be established inside the small pores of those

deformed MNPs. However, when salt concentration is high (1000 mM), EDL already exist in all the small openings of the MNP trees. Either way, variation of the pore size between tree branches makes little change to an extra EDL buildup. In fact, this concentration dependent behavior was briefly mentioned on Page 10. Now we added a short sentence on Page 11: *“this concentration-dependent response can be used to explain the sudden drop for ΔC in Figure 2D.”*

11. Figure 3 C (inset photo). How did the authors control and measure the pressure on the bulb syringe (was this controlled by hand and estimated)? More generally, the use of air pressure to analyze device behavior is not motivated in the paper.

Airflow from the bulb syringe was controlled manually by controlling the extent of bulb deformation in a fixed period of time. While we can read the pressure by placing the syringe close to a digital balance, we use airflow as the main parameter for Figure 3C.

The use of airflow to analyze device behavior is to illustrate shear force can induce a capacitance decrease. In contrast, hydrogel samples back in Table 1 and Figure 2D are applied with compressive forces. We first saw this capacitance increase in Figure 3B, Figure 3C is then used to confirm the capacitance decrease behavior is indeed due to a shear load.

We now added one motivation at the end of the shear force discussion on Page 11: *“Overall, while compressive forces induce capacitance increase for our device, we found shear force can indeed induce a capacitance decrease.”*

12. DOI: 10.1021/ja7106178; "Relation between the Ion Size and Pore Size for an Electric Double-Layer Capacitor"; this paper could be a useful reference to discuss the suggested mechanism for capacitance change in the electrical double layer capacitors.

Thank you for your suggestion. We have read this paper carefully and added it as an additional reference #31 to indicate regulation in pore size could potentially change the capacitance in an ionic conductor. On Page 3, we placed *“...and are particularly sensitive to the size matching between ions and their hosting porous electrode³¹”* at the end of the paragraph.

13. The description of the static pressure tests could be clarified with a schematic (maybe include that in the supplementary section).

We drew a scheme below to clarify the pressure mapping tests and used it as Figure S4 in Supplementary Information.

Figure S4. Schematic of the pressure mapping test with a 4×4 sensing array. Inset shows the hydrogel sample used for this experiment. Briefly, a solid weight (PDMS block) with a

controlled mass and bottom area is placed on a predefined location. Capacitance readings from different paired electrodes (one from the top and another from the bottom) are then recorded to map out the pressure zone on top of the hydrogel sensor array.

14. Throughout the paper, the authors are using different ion concentrations. It would be useful to include typical ion concentrations in the ocean, as this device is intended for underwater audio detection.

When we soak the device in water for acoustic measurements, we wrapped a thin layer of plastic membrane outside to avoid direct contact of the electrodes to the water. Hence, electrical shorting with the conducting media will not occur. While higher concentration of salt can beef up the detection signal of our microphone, it has an undesired harmful effect, i.e., corrosion of the electrodes. Anyhow, we did change the salt concentration inside the hydrogel sample to 600 mM (salt concentration in ocean) and we obtained data on the right. Generally, at a low frequency of 20 Hz, the signal intensity is around -150 dB, but at the high frequency side (3 kHz), the noise level is around -185 dB, with both of these value slightly better than 100 mM signal in Figure 4D. We added this as a new figure in Supplementary Information (Figure S7).

15. The main message of this paper is focused on a new idea to fabricate a high-end audio device. This message is not supported convincingly enough with acoustic experiments and demonstrations. An additional experiment could be the creation of a spectrogram (see ref 28 of the paper): audio input in the form of a sine sweep with constant amplitude from low frequency to high frequency. The signal recorded with the new microphone should be analyzed for frequency content by Fourier transform analysis. This would give important additional info about the quality of the recording, such as: Do the complex micro-architecture of the microphone and the suggested physical mechanism introduce any kind of higher harmonics, increased sensitivity to specific frequencies, resonances, etc.?

We finished the suggested spectrogram experiments. Two spectrograms are shown in the figures here: the one at the top is recorded by the commercial hydrophone and the one at the bottom is recorded by our hydrogel sensor. First we note the high similarity between the two

Commercial Hydrophone:

Hydrogel sensor:

spectrograms, which supports our performance claim. We see the prominent main sweeping lines from bottom left corner to the upper right corner. There are other features that indicate distortion. However, none of these features are from our hydrogel sensor. The vertical lines are artifacts due to oscilloscope not able to save ~15 s scan in one file. It divided the whole scan into 32 files, and between each file some data is lost. Therefore, there is a discontinuity (similar to a step function) of saved signal, and the Fourier transform of that is a wideband vertical line seen on the spectrogram. There are 31 lines equally spaced. The second feature is the harmonics (i.e. parallel lines in addition to the major frequency sweeping line). Since it also appears in the commercial hydrophone measurement in exactly the same pattern, we think these come from the sound generator. In fact, we are currently using a commercial dynamic speaker pushed against a plastic water tank to do the experiment, and the coupling between the two are non-ideal. However, none of these problems are from hydrogel sensor itself and we think it is truly an outstanding underwater acoustic detector.

16. The central message of the paper (extremely sensitive hydrogel microphone) would be strongly supported by providing an audio sample of recordings performed with the hydrogel microphone. How about playing a piece of music or part of a speech over the loudspeaker and providing the recorded sound file in the supplementary materials? Of course, the cut-off at higher frequencies could be a problem, but 3 kHz should be high enough to generate a meaningful result. In any case, this type of data would give readers a much more intuitive feel about the performance of this device.

We performed the recording and the files are now in supplementary materials. We can clearly distinguish all the speech. Compared to the original sound, we also clearly noticed some additional white noise (i.e. the “hiss”). Considering we did not use any low-noise pre-amp or strict EM shielding from the environment, we consider the noise level acceptable. Of course further development would be nice to get high-fidelity and low-noise recording, but that is beyond the scope of this work.

To clarify, we added the following paragraph at the end of the Methods section: *“**Spectrogram and Audio Recording.** Audio input in the form of a sine sweep with constant amplitude from 20 Hz to 3 kHz is applied to the MNP-hydrogel microphone. The signal recorded is analyzed for frequency content by Fourier transform analysis (see Supplementary Information). A short speech is also recorded by using this MNP-hydrogel microphone as part of the Supplementary Information, without using any low-noise pre-amp or strict EM shielding from the environment”*

Reviewer #2 (Remarks to the Author):

The authors have reported a capacitive sensor by using fractal-structured metal nanoparticles on hydrogel matrices and showed the ability of signal sensing under mechanical stimuli in three cases; airflow, static loads and underwater acoustic waves. They utilized the feature of large capacitance from hydrogels enabled by electrical double layer and invented the method for fabricating deformable Ag fractal networks in intended patterns. Their suggestion on the application of their devices as hydrogel microphones underwater is quite interesting but there are some unclear factors I would like to ask and minor points which should be revised. Please consult the followings:

1. In page 12, the absolute sensitivity of the MNP-hydrogel device seems better than conventional dielectric capacitors but the relative sensitivity, which would be more appropriate to be an evaluation criterion of sensing, does not. This makes the description of "extreme sensitivity" in the title unsuitable.

We removed the "extreme sensitivity" from the title and modified it as "*Hydrogel Microphone: Deformable Network of Metal Nanoparticles Enables Stealthy Underwater Listening*".

2. The overall accounts on Figure 4D are vague. There should be more specific descriptions, especially on the high frequency regions. In addition, when the frequency follows a cyclic pattern, does the MNP-device maintain identical detection behaviors? Is there any chance to show a hysteresis?

We added the following description to figure caption of Figure 4D: "*Ion concentration and bias direction affect the performance of the hydrogel microphone, where a higher ion concentration (100 mM) under a positive bias (copper positive, ITO negative) gave the best result (-152 dB@20 Hz). Device performance towards a high frequency sound (3 kHz) is however limited, with a response around the noise level of -190 ~ -200 dB. This is mainly caused by the slow motion of ions.*"

As for cyclic loads, we swept the frequency with a network analyzer. It generates sine waves from 20 Hz to 3 kHz during a period of 15 s and then repeated thereafter. Backward scan is also performed. For all the samples we tested, we have not observed any hysteresis; neither do we see large fluctuations. All the responses are very similar. Standard deviations for multiple consecutive tests were included in Figure 4D even though the small value of them makes the error bars hardly noticeable.

3. In Figure 3A, the illustration and its caption can be misunderstood - Since the distance of EDL, which is in atomic size, does not affected by external stimuli (Conway B.E., et al., Modern Aspects of Electrochemistry v.31, Kluwer, 1999), there should be no variation in gaps of EDL. The explanation on capacitance variations by external stimuli needs to be revised more clearly.

Electrical double layer consists of two parts: one is the compact layer which closely sits next to the electrode surface, with a gap distance (δ) of subnanometer, the other is the diffusion layer caused by irregular thermal motion of ions, which can extend to some distance away from the electrode surface. Notably, the thickness of this diffusion layer (d) is related to Debye length (κ^{-1}), which can increase from subnanometers to several hundreds of nanometers with a

decreasing ion concentration. Collectively, the total capacitance of EDL can be regarded as two capacitors connected in series, one is compact layer-based (C_δ) and the other one diffusion layer-based (C_d), following a relationship of $\frac{1}{C} = \frac{1}{C_\delta} + \frac{1}{C_d}$. In our case, the electrolyte solution is at low molarity, where the Debye length $\kappa^{-1} \gg$ compact layer gap distance δ . The EDL capacitance is mainly determined by C_d and the value can be expressed as: $C \approx \frac{\epsilon_0 \epsilon_r A}{d}$, here d is usually tens to hundreds of nanometers. As seen from the SEM image (Figure 2C), the rooms between Ag branches are in fact having a size in this range. Therefore, it is reasonable to assume, when the EDL thickness is comparable to these pore sizes, EDL formed on the inner walls of the pores will overlap, resulting in ion exclusions or the surface areas of these pores cannot be efficiently used to adsorb ions. However, when these rooms become larger under external pressures, buildup of new EDL becomes possible. Discussions of Debye length and the diffusion based EDL can be seen in multiple references #30, 32, 53 and 54, where #53 and 54 are newly added to the reference of the paper.

In the main text we also explained this more clearly (Page 8): “*Instead of being a uniform structure, EDL consists of two parts^{30,32,53,54}: one is the compact layer which closely sits next to the electrode surface, with a gap distance (δ) of subnanometer, the other is the diffusion layer caused by irregular thermal motion of ions, which can extend to some distance away from the electrode surface. Notably, the thickness of this diffusion layer (d) is related to Debye length (κ^{-1}), which can increase from subnanometers to several hundreds of nanometers with a decreasing ion concentration³⁰. In our case, the electrolyte solution is at low molarity, where the Debye length $\kappa^{-1} \gg$ compact layer gap distance δ . Hence, the EDL capacitance is dominated by the diffusion layer where the thickness is usually tens to hundreds of nanometers. Certainly, when salt concentration jumped from 10^{-2} to 1000 mM, the hydrogel capacitor delivered more than 300 times increase in capacitance.*”

4. Is there any relationship between the sensitivity of the device and mechanical properties of hydrogel matrices?

In principle, the mechanical properties of hydrogel will affect the deformation of the metal nanoparticle network, which should affect the device sensitivity. However, experimentally embedding metal network into hydrogel is a multi-step chemical process and it is nearly impossible to only change the hydrogel properties without changing the metal network. Therefore we are not able to isolate the effect of hydrogel mechanical properties experimentally.

However, we predict the sensitivity in air will be much more dependent on the matrix than that device tested in water. Mainly this is due to a high acoustic coupling of the gel membrane to underwater sound.

5. There is no information on input signals for detecting underwater sound in Figure 1D. Input signals are expected to be added on the graph for comparison with the output.

Thanks for this suggestion. For Figure 1D the input signal is simply a sine wave, however it is distorted due to the non-ideal loudspeaker-water tank coupling. Therefore, it is very difficult to know the direct acoustic input signal on the hydrogel. To explain this coupling issue, we now added spectrogram of frequency sweeping experiments in the supplementary materials. Two spectrograms are shown in the figures here: the one at the top is recorded by the commercial

hydrophone and the one at the bottom is recorded by our hydrogel sensor. First we note the high similarity between the two spectrograms, which supports our performance claim. We see the prominent main sweeping lines from bottom left corner to the upper right corner. There are other features that indicate distortion. However, none of these features are from our hydrogel sensor. The vertical lines are artifacts due to oscilloscope not able to save ~15s scan in one file. It divided the whole scan into 32 files, and between each file some data is lost. Therefore, there is a discontinuity (similar to a step function) of saved signal, and the Fourier transform of that is a wideband vertical line seen on the spectrogram. There are 31 lines equally spaced. The second feature is the harmonics (i.e. parallel lines in addition to the major frequency sweeping line). Since it also appears in the commercial hydrophone measurement in exactly the same pattern, we think these comes from the sound generator. In fact, we are currently using a commercial dynamic speaker pushed against a plastic water tank to do the experiment, and the coupling between the two are non-ideal. However, none of these problems are from hydrogel sensor itself and we think it is truly an outstanding under water acoustic detector.

Commercial Hydrophone:

Hydrogel sensor:

6. In page 12, authors have mentioned an additional factor on the variation of EDL capacitance by ion concentration modulation (excluding mechanical stimuli) - the fact that MNPs are electrically biased. Is there any complementary explanation on the effect of electrical bias on the MNP-hydrogel device?

The influence of electrical bias to MNP-hydrogel is coupled with the deformation of the MNP network. When MNP deforms, electrical bias will act as a driving force to attract or repel layer of ions. As such, drift of ions throughout the hydrogel membrane is affected by the electric field. We provided a detailed analysis in Supplementary Information to explain the ionic waves.

7. How did you measure the sound pressure in Figure 4C?

We used a commercial listening device (hydrophone, SQ 26-07) with known sensitivity to calibrate the sound pressure in our experiments. In a typical process, the hydrogel sample was replaced by hydrophone at the same location, and then the same sound source was applied. Through the response signal from the hydrophone, sound pressure at this location can be calculated. Since sensitivity (S) of hydrophone is giving by:

$$S = 20 \log_{10} \frac{V_{RMS}}{V_0}$$

Where S has a value of -169 under a sound source of 100 Hz and V_0 is the reference voltage where 1.0 V refers to 1 μ Pa (Provided by the device company: Cetacean Research Technology, Seattle, WA). V_{RMS} is the root-mean-square of response signal and can be calculated by converting the peak-to-peak voltage (V_{pp}) recorded by oscilloscope:

$$V_{RMS} = \frac{V_{pp}}{2\sqrt{2}}$$

So based on the known sensitivity (S) and V_{RMS} , V_0 can be then calculated. Since each 1.0 V of V_0 refers to 1 μ Pa of pressure, the value of V_0 will then tell us the sound pressure generated by the sound source at that location. More details can be found in the supporting information.

8. The word "fractal structure" is confusing. Please consider an alternative.

“Fractal structure” was used to describe those dendritic metal networks. We changed the term into “continuous network” or “sponge structure” throughout the manuscript.

9. At the 6th line in page 4, I cannot catch up with the meaning of "figure 8."

Our hydrogel microphone can respond to sound from different directions, with maximum sensitivity to those from normal directions. The form of “8” in shape is frequently used to illustrate the directional response of sound wave detecting devices. To make it clear, in manuscript (page 4, line 6), we replaced the original description by “*with the directional response (1D right) having a maximum value when sound incidence is perpendicular to the surface of the hydrogel microphone but a much weaker one when sound is parallel to the microphone surface or a typical “figure 8”-like pattern*”.

10. Schematics in Figure 3B and 3C (especially the inset photo) are not effective to deliver the information.

To make Figure 3B easier for reading, we added two side arrows and a legend ($\Delta C/C_0$) in the figure. We also modified the description for 3B.

Figure 3. (B) Array of MNP in hydrogel can be used to map out contour of shaped loads of 8.0 kPa, from left to right for the shape of an “O”, an “L”, and a small dot. Capacitance increases (green; arrow up) or decreases (red; arrow down) due to location of the loads.

Besides, we drew an additional scheme as in the Supplementary Information to explain the experimental setup.

Figure S5. Schematic of the pressure mapping test with a 4×4 sensing array. Inset shows the hydrogel sample used for this experiment. Briefly, a solid weight (PDMS block) with a controlled mass and bottom area is placed on a predefined location. Capacitance readings from different paired electrodes (one from the top and another from the bottom) are then recorded to map out the pressure zone on top of the hydrogel sensor array.

11. In the manuscript for Figure 2D, the explanation is unclear and notations in Figure 2D are not eye-catching. There should be some modifications.

Data plotted in Figure 2D are those from Table 1. We illustrated this by adding a sentence in the caption of Figure 2D: “Note: Data plotted here can be found in Table 1, where statistic measurements using multiple samples or the same sample with multiple tests are used.” We also modified those unfilled square or circle data points in this figure with color filled squares or circles, enlarged the marker size from 5 to 7, and shifted the arrow indicator on the right bottom corner to enhance readability.

12. It would be better to adding a legend in Figure 4C.

We added two legends respectively for those marks with a shape of square or circle: (Circle) 10^{-2} mM; (Square) 100 mM.

Reviewer #3 (Remarks to the Author):

Evaluation of Hydrogel Microphone: Deformable Network of Metal Nanoparticles Enables Stealthy Underwater Listening with Extreme Sensitivity

Authors Yang Gao et al

The authors describe a microphone that combines hydrogel with metal particles, where sound on system supposedly causes capacitance changes. The claim is that the good impedance match of water to hydrogel makes the microphone stealthy (this term even appears in title of paper). I find that while the work has some interesting features, it is highly overblown and not suitable for publication in this journal.

1. The authors assert that this microphone is the only obstacle to make submarines totally undetectable. This is laughable! Either the authors do not understand how acoustic or EM cloaking really works, and they do not appreciate the complexities of sound suppression of submarine propulsion, or they are intentionally misleading reader.

Several ways can detect a submarine, one by passively listening the noise produced by the submarine propeller, and another by the acoustic reflection from the submarine by an active SONAR scanning. While efforts are made to reduce the propeller noise, other efforts such as acoustic cloaking are used to reduce the reflected acoustic waves. Our intention is placed on the latter and this is indicated by our manuscript title as "...stealthy underwater listening". In brief, we are stating the fact that our sensor does not reflect any acoustic waves under SONAR scanning.

To avoid confusions to readers, we modified the sentence "*Nowadays with an acoustic metamaterial cloak designed to conceal submarine bodies from SONAR detection, the piezoelectric detectors or cavity-based microphones outside of this "invisibility cloak" becomes the only obstacle to make submarines totally undetectable*" on Page 2 to "*Nowadays with an acoustic metamaterial cloak designed to conceal submarine bodies from SONAR detection and active efforts to reduce noise emission from the submarine itself, the piezoelectric detector or cavity-based microphone outside of this "invisibility cloak" becomes an undesired acoustic reflector under active SONAR scanning*" and deleted the redundant claim prior to this sentence. We hope this eases the concern from the reviewer and gives readers an unbiased view.

2. Second, although the configuration presented is said to be capacitive, there are strong similarities to conventional carbon pack microphone, where small carbon particles are loosely packed (often with no direct acoustic chamber) such that acoustic wave impinging causes particles to change connectivity (and capacitance!). That microphone invented in the 1870s was the basis of the telephone industry for over 100 years! Why no reference to this?

We did not include this work during our initial writing as this carbon microphone operates as a tunable resistor, instead of a capacitor. Before the end of the Introduction on Page 4, we added a new statement to acknowledge this pioneering work from David Hughes (also cited his patent as ref. #36): "*Unlike David Hughes' early version of acoustic transmitter³⁶ (carbon microphone) that works by compacting loosely connected carbon particles between two metallic plates with a change in resistance, our device is completely void or cavity-free, implying zero-reflection toward SONAR scanning.*"

3. The authors compare impedance mismatch of hard piezo material to hydrogel. But there are many non-piezo microphone systems around. Even humans can hear really well underwater (in fact better than in air, that is why they lose stereo direction information). And they don't use piezos either.

We included non-piezo based mechanisms like sense from fish, vibration from graphene, or even dielectric capacitors as listening pathways in our introduction. In comparison to all these pathways, our device functions with a new mechanism that was not reported before; it also reveals an unmatched performance beyond stealthy listening, particularly for low frequency acoustic waves.

4. This device seems to be sensitive to ion infiltration. So fluctuations in salinity (example) give different and likely uncontrolled response. This work is best for a journal other than Nature Communications.

We used a plastic wrap to insulate the hydrogel microphone from its aqueous environment. To clarify, we changed the sentence on Page 12 from “...soaked the MNP-hydrogel device in water and...” to “...sealed the MNP-hydrogel device with a plastic wrap, soaked it in water and...”. We also added one sentence on Page 20: “A plastic wrap is used to insulate the hydrogel device from its aqueous environment, therefore no current shorting or ion infiltration between the device and its liquid background occurs. Then...”

Response to reviewer 1:

1. Although some areas of the paper have improved, quality of writing is still below the level I would expect for this journal. Sentences are over complicated and it is sometimes difficult to follow the writing. Maybe have a native speaker revise the article.

Thanks for this suggestion. Our co-author, Professor Stephen Ducharme, has carefully revised the writing of the manuscript. All these revisions are made in the manuscript file using 'track changes', and we also provide another version with all the revisions marked in YELLOW.

2. Use of statistics in Table 1 (and throughout the manuscript): Different samples, how many, and how large were the experimental errors? For Table 1, according to authors "we used two ways to check repetitivity or variability of our hydrogel samples" one by using multiple samples (3 or more) and with the same sample but 3 times of data recording. Here, I don't think the following explanation is clear enough: "rather constant value in relative standard deviation (standard deviation over average)". Also, where do the values for standard deviation come from? To simplify the explanation of the statistics used, could the authors say "3 or more samples were tested multiple times"? Most importantly, the exact statistical analyses that were used throughout the manuscript should be clearly conveyed to the reader. If three samples were used the reader should have access to this information.

Thank you for your suggestions. Raw data are added in Supplementary Table 1.

To make it clearer to the readers, the note below Table 1 is revised as: "*Note: All the values of C_0 , ΔC and $\Delta C/C_0$ showed here are the average values using 3 samples by performing one pressure test on each sample (Supplementary Table 1).*" The error bars are described in the figure caption of Figure 2d: "*(D) MNP-hydrogel (solid lines) responds to a static pressure of 5.4 kPa with more than four times in relative capacitance change or 7-8 times in capacitance change than MNP-free device (dashed lines). Note: Data from Table 1. The error bars for ΔC and $\Delta C/C_0$ in MNP-hydrogel are respectively 21-25% and 3.5-4.5%. For MNP-free hydrogel, the error bars of ΔC and $\Delta C/C_0$ are respectively 13.0-13.5% and 3.0-4.0%.*"

Supplementary Table 1. Statistic measurements of hydrogel capacitors

			Salt concentration (mM)				
			10^{-2}	1	10	100	1000
MNP-free hydrogel	C_0 (nF)	C_{01}	0.94	6.38	15	81.5	312.5
		C_{02}	0.75	7.22	17.27	90.82	365.6
		C_{03}	0.88	5.87	14.32	70.02	284.7
		Average	0.86	6.49	15.53	80.78	320.9
		Standard Deviations	0.097	0.682	1.545	10.419	41.104
		Relative Standard Deviations	0.113	0.105	0.099	0.129	0.128
	ΔC (nF)	ΔC_1	0.052	0.309	0.612	2.11	5.142
		ΔC_2	0.04	0.361	0.732	2.51	6.11
		ΔC_3	0.05	0.278	0.57	1.95	4.63
		Average	0.047	0.316	0.638	2.19	5.294
		Standard Deviations	0.006	0.042	0.084	0.288	0.752
		Relative Standard Deviations (Errors)	0.136	0.133	0.132	0.131	0.142
	$\Delta C/C_0$	$\Delta C_1/C_0$	0.055	0.048	0.041	0.026	0.016
		$\Delta C_2/C_0$	0.053	0.05	0.042	0.028	0.017
		$\Delta C_3/C_0$	0.057	0.047	0.039	0.028	0.016
Average		0.055	0.048	0.041	0.027	0.016	
Standard Deviations		0.002	0.0013	0.0013	0.0011	0.0002	
Relative Standard Deviations (Errors)		0.036	0.027	0.032	0.039	0.0137	
MNP-hydrogel	C_0 (nF)	C_{01}	3.4	15.63	36.67	170.73	476.2
		C_{02}	2.3	19.33	28.84	212.18	395.11
		C_{03}	3.7	12.44	44.02	141.24	562.87
		Average	3.13	15.8	36.51	174.72	478.06
		Standard Deviations	0.737	3.448	7.591	35.638	83.895
		Relative Standard Deviations (Errors)	0.235	0.218	0.208	0.204	0.175
	ΔC (nF)	ΔC_1	0.83	2.52	4.44	14.39	10.11
		ΔC_2	0.52	3.31	3.71	16.75	8.81
		ΔC_3	0.85	2.11	5.68	10.97	13.07
		Average	0.733	2.646	4.61	14.04	10.66
		Standard Deviations	0.185	0.61	0.996	2.906	2.184
		Relative Standard Deviations (Errors)	0.252	0.231	0.216	0.207	0.205

$\Delta C/C_0$	$\Delta C_1/C_0$	0.244	0.161	0.121	0.084	0.021
	$\Delta C_2/C_0$	0.226	0.171	0.128	0.079	0.022
	$\Delta C_3/C_0$	0.23	0.17	0.129	0.078	0.023
	Average	0.233	0.167	0.126	0.08	0.022
	Standard Deviations	0.009	0.005	0.0045	0.0035	0.001
	Relative Standard	0.041	0.033	0.036	0.044	0.045
	Deviations (Errors)					

Note: Relative Standard Deviations = Standard Deviations/Average

3. Figures S5 c) and d) show voltage amplitude from hydrogel for different ion concentrations. The voltage scales are different here. If the vertical scales for each plot from the 10^{-2} and 100 mM were the same (as input signal increases and between different ion concentrations) the effects of increasing input and ion concentration would be clearer.

For the input signal in Supplementary Figures 5c and 5d, we have changed the vertical scales to be the same. For the output signal in Supplementary Figure 6 (in new version, it became Supplementary Figure 4), if we keep the vertical scales the same, the response curves will be flat for those at low input signals. We slightly modified the y-axis to manifest the contrast between signals of different outputs.

Supplementary Figure 5. (a) Response (voltage output) of a commercial hydrophone (SQ 26-07) to a selected reference acoustic wave. (b) Sensitivity map of hydrophone SQ 26-07 (provided by Cetacean Research Technology) towards varieties of frequencies (10-60 kHz). A series of acoustic waves as input signal applied on the

hydrogel microphones with different ion concentrations (c) 10^{-2} and (d) 100 mM, which are recorded by oscilloscope directly. Note: These pulsed signals are from the loudspeaker/amplifier directly, not from the hydrogel sensors.

Supplementary Figure 4. Response (voltage output) of hydrogel microphones with different ion concentration (a) 10^{-2} and (b) 100 mM towards acoustic waves as in Supplementary Figure 5(c-d).

4) The signal shown in figure 1 D is different from a sine wave. The authors explain in their rebuttal letter that the sound transfers from the amplifier and loudspeaker to the water tank and thereby becomes non-linear. It may be helpful to add this information to the paper or supplemental -- it is possible other readers have this same question.

Thank you for your suggestions. We have added this into the manuscript on Page 6, “Moreover, the MNP-hydrogel microphone has a wide frequency response, up to 2 kHz (Figure 1e) and has a pronounced directional sensitivity perpendicular to the sensor surface (Figure 1f). The slight distortion of response signals can be ascribed to the non-ideal sound transfers from the amplifier and loudspeaker to the water tank (Supplementary Figure 1).”

5) The authors explain in their rebuttal letter that airflow from the bulb syringe is controlled by timing how long it takes until the bulb is evacuated. This information should be added to the paper in the methods section.

Thank you for your suggestions. We have added this in the methods section (Static Pressure Measurement): “The airflow was generated by gently squeezing a rubber blower fixed on a ring stand, which the extent of bulb deformation was executed over a fixed period of time, with the pressure being read by placing the syringe close to a digital balance. The angle of blowing is adjusted to ensure the tip of the blower

always pointing to the center of the Cu electrode. The distance between the tip of rubber blower and the surface of hydrogel was fixed at 10 mm, and the pressure generated by gripping was controlled at 1.0 kPa.”